

# The Spatial-Temporal Evolution of the Chongzhen Drought (1627-1644) in China and its Impact on Famine

Siying Chen[1,2], Yun Su[1*], Xudong Chen[1], Liang Emlyn Yang[2]

Faculty of Geographical Science, Beijing Normal University, Beijing 100875, PR China

2 Department of Geography, Ludwig Maximilian University of Munich (LMU), Munich 80333, Germany

*Correspondence to*: Yun Su (suyun@bnu.edu.cn)

**ABSTRACT.** Investigations of past extreme climate events offer insights into the interactions between natural forces, ecosystems, and human societies. The Chongzhen Drought, which occurred from 1627 to 1644 CE, stands as possibly the most severe drought in central and eastern China over the last 1500 years, remarkable for its duration, extent, and the vast number

of people affected. Concurrently, a widespread famine emerged, triggering peasant uprisings that were argued as having contributed to the Ming Dynasty's downfall. This study extracted 1,802 drought records and 1,977 famine records from Chinese historical documents to reconstruct the spatial-temporal progression of the drought and its impact on famine. The records provided drought information of season, duration, and intensity, which enabled a classification of four drought severity levels. Then kernel density estimation reconstructed the spatial pattern of drought and the Drought Kernel Density Index (DKDI)

series in sub-regions. Between 1627 and 1644, the drought affected most of central and eastern China. The severe drought zone was mainly located north of 29°N, shifting from Northwest China to North China and then expanding to the south. The development of drought in different regions was not synchronized. Northwest China faced the earliest drought outbreak, which eased in the middle period and peaked in 1640. North China's DKDI series forms a single-peaked curve, indicating a gradual aggravation of the drought from 1633 to 1640. The Yangtze-Huai Region's DKDI series shows a multi-peaked curve, with

repeated cycles of worsening and easing drought, peaking in 1641. Furthermore, the reconstruction of the spatio-temporal progression of famine revealed overlapped ranges and similar development trends to that of the drought. The years marking the peak of the Famine Kernel Density Index (FKDI) in Northwest China, North China, and the Yangtze-Huai Region coincided with those of the DKDI. Regression analysis identified drought as the primary factor triggering famine, accounting for approximately 67.3% of its occurrence. In North China, the contribution of drought was higher, at 73.4%. Series and

correlation analyses indicate a continuity of 2-3 years in drought's impacts on famine. The paper further clarified the dominant pathway of climate impact transmission in this case: extreme drought → declining agricultural harvest → food shortage → famine. Other natural and socio-economic factors, such as locust infestations, nomadic invasions, and economic decline, also played a role in the occurrence of famine. Human response measures were instrumental in regulating the transmission of climate change impacts.



# 1 Introduction

Climate change, as the most dynamic component of the earth system, serves as a crucial backdrop for the evolution of human civilization. Numerous studies have highlighted the profound impacts of climate change on agricultural production (Wright and Thorpe, 2003; Yin et al., 2015), human health (Anthony et al., 2006), migration (Büntgen et al., 2016), economic development (Pei et al., 2013), social rise and fall (Pederson et al., 2014;Yancheva et al., 2007; Zhang and Lu, 2007; Yin et al., 2016), and the collapse of civilizations (Harvey and Raymond, 2001; Cullen et al.,2000; Buckley et al., 2010; Haug et al., 2003). The interaction between climate change and human societies on different time scales represents a vital research area of global change. Investigations into significant historical extreme weather events and their societal impacts are instrumental in understanding such interactions (PAGES, 2009; Yang, et al., 2019).

Climate change could reduce food security and exacerbate poverty (IPCC, 2022). Famine refers to a condition where widespread hunger occurs due to a lack of food, indicating a failure to meet basic survival and health needs at the individual level and a crisis in food security at the societal level. From a natural perspective, extreme weather events affect agricultural harvests, leading to food shortages that may directly trigger famine. Drought, in particular, has frequently precipitated famine throughout Chinese history (Teng et al., 2014). Climate change on longer time scales can alter the area of arable land, crop structure, growing periods, etc. (Zhang, et al., 2021; Chen, et al., 2021), thereby increasing or decreasing the likelihood of famine. As for social consequences, famine was closely related to displacement, plague, and social unrest. Historically, the large-scale peasantry uprisings in China in the late Eastern Han, Western Jin, late Sui, late Tang, late Ming, and late Qing dynasties all erupted in the context of extreme drought and famines (Fang et al., 2019). Famine is a significant manifestation of the adverse effects of climate change reaching the human system. It also serves as a vital link in the chain of transmission of these effects to the economic, political, and military domains, which is particularly evident in agrarian societies.

1627-1644 CE saw an extraordinary and extreme drought in China, known as the "Chongzhen Megadrought" because it coincided with the last period of the Ming Dynasty, the Chongzhen Emperor's reign. This drought has been corroborated by natural evidence of stalagmites, tree rings, sporopollenin, and historical documents (Zhang et al. 2008; Tan et al. 2011a; Fang et al. 2012; Zhang et al. 2017). This period is characterized by weak monsoon activity and a dry climate, likely influenced by phenomena such as El Niño events and violent volcanic eruptions (Shen et al. 2007; Liu et al. 2014). The Chongzhen Megadrought is potentially the most severe drought in eastern China during the past 1500 years, and its duration and geographical extent, as well as the number of affected people, were rare in history (Tan 2003; Zheng et al. 2006; Peng and Xu 2009). The drought led to a decline in food production, which triggered a widespread and enduring famine, manifesting as food shortages, food substitution, migration, starving to death, and the breakdown of social order. A large number of famine victims joined the peasant uprising army, which played a pivotal role in the collapse of the Ming Dynasty (Zheng et al., 2014; Fang et al., 2019). The progression of the Peasants' War was also closely related to the development of the famine (Cao, 2019).

Several studies have examined the duration, extent, and progression of the Chongzhen drought (Chinese Academy of Meteorological Sciences, 1981; Chen, 1991; Fang, 2006; Liu et al., 2014; Tan, 2001; Guo, 2014), yet many of these lacked detailed spatial and temporal resolution, or they concentrated solely on limited areas. There is still a shortfall in research offering yearly temporal resolution for drought reconstruction, and at the same time, encompasses the entire affected region. Regarding the impacts of the Chongzhen Drought, existing research has mainly focused on historical accounts, particularly highlighting the drought as a catalyst for peasantry uprising (Cao, 2019; Chen, 2006; Zheng et al., 2014; Wang et al., 2010; Chen and Zhu, 2003; Li, 1999). However, the specific processes and mechanisms through which drought impacts were transmitted to the human system remain poorly understood, with an absence of quantitative or semi-quantitative analyses.

In the case of the Chongzhen Drought, the drought represented a manifestation of the natural system, whereas famine emerged as a typical manifestation of the human system. This study, therefore, concentrates on these two critical aspects to





investigate the transmission of extreme climate impacts. We extracted drought records from Chinese historical documents from 1627 to 1644 and divided them into different levels based on semantic differences. To illustrate the spatial and temporal development, we employed kernel density estimation, creating a year-by-year spatial pattern of drought and the series of

75    Drought Kernel Density Index (DKDI) across various regions. A similar approach was taken for famine records. By comparing the spatial-temporal patterns of drought and famine and conducting statistical analysis, we investigate their interconnection and how drought precipitates famine. The study area is the region of China with recorded droughts during this period (the colored portion in Fig. 1), which was further divided into five subregions based on physical geography and socio-economic distinctions.

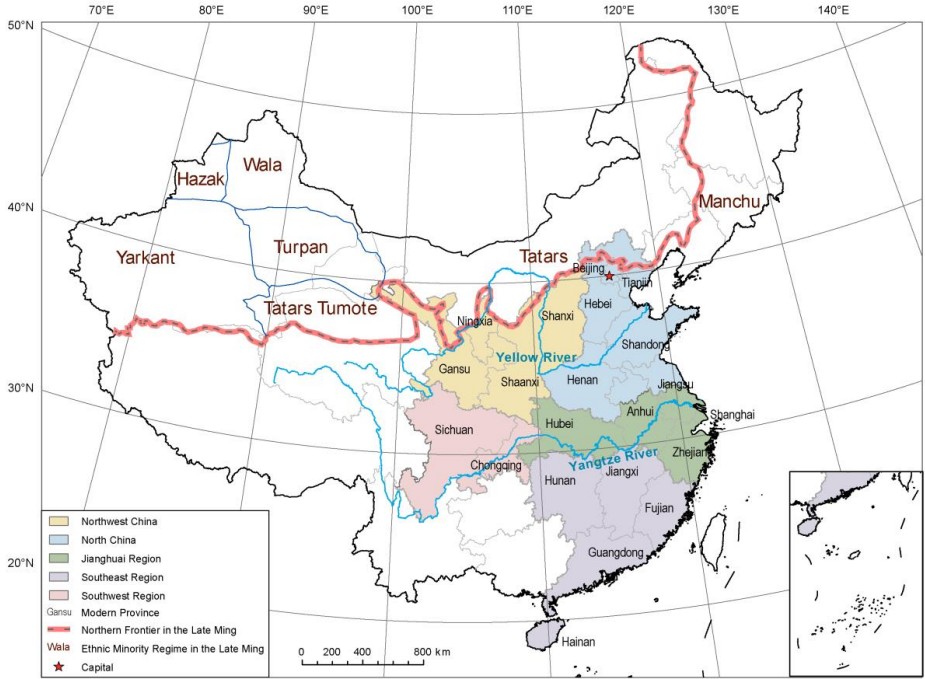

**Figure 1: Map of the study area and subregions in colors**

## 2 Data and methods

### 2.1 Data source

Historical documents are invaluable proxies for reconstructing past climate change, with the advantages of high
85    temporal resolution, location accuracy, and clarity of climatic significance (Pfister, 2008). China owns a wealth of historical documents lasting for thousands of years without interruption. In recent decades, meteorological departments and scholars have mined these documents for climate and disaster-related records, assembling them into comprehensive historical data collections. Records related to disasters within the study area during 1627-1644 were extracted from 24 collections (see Appendix A). The records were organized in six fields: year, location, province, content, source, and original document

90    provenance, thereby establishing a detailed database of disasters in east-central China from 1627 to 1644. The "year" field was standardized to the Common Era notation of the Gregorian calendar. The "Location" was updated to reflect current place names, guided by *the Historical Atlas of China* (Tan, 1982), with counties serving as the basic geographic administrative units. For the "province" field, the current administrative divisions of the People's Republic of China were applied. The



"content" is the text description regarding disasters. The "source" refers to the name of the collection where the record came from. The "original document provenance" mainly includes local chronicles, *the History of Ming*, and *Ming Shilu*. Records sharing the same year and location were merged, resulting in a database comprising 6,282 entries.

### 2.2 Processing of drought records

All records directly related to drought were extracted from the database, totalling 1,802 entries. The contents of these drought records can be divided into four categories: (1) Precipitation, which often includes details about the season or month, e.g., "late spring rains", "no rain in June"; (2) Hydrology concerning rivers, lakes, springs, and groundwater, e.g., "the river is shallow enough to walk through", "wells dry up", "the lake's bottom is bursting"; (3) Plant growth, e.g., "the grass is withered", "no grass in the wild"; and (4) Comprehensive descriptions of drought conditions, e.g., "drought", "severe drought". To study the relationship between drought and famine, it is crucial to maintain the independence of these two sets of data. Therefore, drought records were extracted and graded based on the manifestations in natural systems—such as the atmosphere, hydrosphere, and biosphere—while deliberately excluding records on the social damages and losses caused by drought.

Based on semantic differences, we distinguish the meanings of qualitative text within the historical records to categorize drought events into various levels and thus measure drought. It is worth noting that the study area is characterized by a typical monsoon climate, with precipitation highly concentrated in the summer and autumn. Consequently, droughts occurring in the wet seasons (summer and autumn) are indicative of greater precipitation anomalies compared to those in the dry seasons (spring and winter). The criteria for rating droughts were established by considering the season of occurrence, duration, and intensity of drought events (Table 1).

Table 1: Criteria for classifying drought levels

| | Criteria | Keyword Examples |
|---|---|---|
| Level-1 | 1) Drought in the dry season and no drought in wet season | Winter drought/ great drought/ no rain in winter (冬旱/冬大旱/冬无雨). Spring drought/ great drought/ no rain in spring (春旱/春大旱/春无雨) |
| | 2) One-month drought | Drought/ great drought/ no rain in a certain month (某月旱/大旱/无雨) |
| Level-2 | 1) Drought in one wet season | Summer drought (夏旱). Autumn drought (秋旱). |
| | 2) "Drought" occurred in a certain year is recorded (without accurate seasons or months) | Drought in the third year of Chongzhen period (崇祯三年旱). |
| | 3) Plant wilting | The grass and trees are withered and scorched (草木为焦). |
| | 4) The river and lake levels have dropped significantly, but have not dried up completely | The river is shallow and walkable (河水浅可步). The rivers run dry with only a small stream remaining (江流涸如带). |
| Level-3 | 1) Drought in two wet seasons, or severe drought in one wet season | Summer-autumn drought (夏秋旱). Great summer drought/ no rain in the summer (夏大旱/夏无雨). Great autumn drought/no rain in the autumn (秋大旱/秋无雨). |
| | 2) "Great drought" occurred in a certain year is recorded (without accurate seasons or months) | Great drought in the tenth year of Chongzhen period (崇祯十年旱) |
| | 3) Complete drying up or ceasing to flow of rivers and lakes, or lowering of groundwater levels | Drying up of wells (井泉竭). Long rivers cease to flow (长河断流). River bottoms are cracked (河底皆龟裂). No springs from wells (凿井不得泉) |
| | 4) Extensive plant mortality | No grass in the wild (野无青草). Without the slightest bit of grass |





| | | growing (寸草不生). |
|---|---|---|
| Level-4 | 1) Drought throughout the year, or severe drought in two wet seasons | Year-round drought/ great drought/ no rain in the whole year (全年旱/大旱/无雨). Drought in all months (十二月俱旱). Great drought/ no rain in summer and autumn (夏秋大旱/夏秋无雨). |

All drought records were rated according to Table 1. Finally, the classification yielded 210 records for Level 1, 631

records for Level 2, 905 records for Level 3, and 56 records for Level 4.

**2.3 Processing of famine records**

A total of 1977 records directly related to famine were extracted from the database. Famine records were rated

according to the criteria in Table 2, yielding 430 records for Level 1, 791 records for Level 2, and 756 records for Level 3.

**Table 2: Criteria for classifying famine levels**

| | Criteria | Keyword Examples |
|---|---|---|
| Level-1 | **Mild famine**: food shortages, but can be sustained by scrimping or receiving relief; sporadic population migrations. | Famine/ Hunger (饥/荒/饉/祲). Relief (赈). Tax exemption (蠲). Hard to get enough food (民艰于食). Eat chaff (食豆饼糟糠). Homeless people on the road (路有流民). The price of rice became expensive (米价涌贵). |
| Level-2 | **Moderate famine**: extreme food shortages, affecting all populations in the region; search for alternative foods; large-scale migration and sporadic deaths; social disorder. | Great famine/ Great hunger (大饥/大荒/大祲). Eat bark/ grass roots/ soil (人食树皮/草根/观音土). The roads are full of displaced people (流亡载道). Nine out of ten houses were empty (十室九空). Occasionally dead people (间有死者). Starving people in the wild (野有饿莩). Thieves rising (盗蜂起). Robberies (聚众抢夺). Selling children (卖儿鬻女). People's livelihood is difficult (民不聊生). |
| Level-3 | **Severe famine**: extremely severe disaster and famine resulting in large numbers of deaths; cannibalism; breakdown of morality, ethics, and social order. | Cannibalism (人相食). Countless people starved to death (饿死者无算). Thousands of dead (死者数万). Roads strewn with overlapping corpses (死者枕藉). Four or five out of ten starved to death (饿死者十之四五). An unprecedented famine (亘古奇荒). |

Note: the criteria were adapted from (Xiao, 2020) and (Wei, 2020).

**2.4 Methods for reconstructing the spatial-temporal progression of drought and famine**

Kernel Density Estimation (KDE) is a non-parametric method employed in spatial analysis to delineate trends in the

spatial distribution of discrete point elements (Wang, 2006). This technique was utilized to reconstruct the spatial patterns of

drought on an annual basis, as well as cumulatively, for the period 1627-1644. The locations of drought occurrences were

identified and represented as point elements. These points were subsequently analyzed using the KDE tool, with the drought

level as the attribute value (termed the "population" field). Given that the basic spatial unit of the drought records was the

county, a search radius of 65 km was selected, which equals the mean distance between adjacent administrative counties

during the late Ming Dynasty (Zhou, 2007). The value of kernel density for a given raster is influenced by the quantity of

surrounding drought events and their respective levels. Consequently, it can be inferred that a higher kernel density value

signifies a more severe drought condition within the area. Employing the same methodology, the spatial distribution of

famine was also reconstructed.

To examine the evolution of drought, the Drought Kernel Density Index (DKDI) series for each region was

reconstructed with an annual temporal resolution. Initially, the average drought kernel density value for each region was



calculated based on the kernel density distribution map. The results were then normalized to obtain DKDI values. The

normalization was conducted by the following equation:

$$DKDI_{ij} = \frac{d_{ij} - min(d)}{max(d) - min(d)}$$

where $DKDI_{ij}$ is the Drought Kernel Density Index for region $j$ in year $i$, $d_{ij}$ is the mean value of drought kernel density

for region $j$ in year $i$, $min(d)$ is the minimum value of the annual kernel density mean for each region, and $max(d)$ is

the maximum value of the annual kernel density mean for each region. Thus, the DKDI series for the period 1627-1644 were

obtained for each region. The same method was employed to reconstruct the Famine Kernel Density Index (FKDI) series.

Compare the spatial patterns and temporal series of drought and famine to investigate their relationship. Statistical

analyses, including Spearman correlation and regression analysis, were conducted on the DKDI and FKDI to examine the

contribution of drought to famine.

**3 The evolution of drought**

The Chongzhen Drought is characterized by its extensive spatial coverage and prolonged duration. Figure 2a presents

the general distribution of drought from 1627-1644. The drought affected a vast area across central and eastern China. The

majority of the severe drought zone appeared north of 29°N, mainly consisting of three parts: (1) the area encompassing

Beijing, southern Hebei, northern Henan, and western Shandong, which was the political heart and seat of the capital of

China at the time; (2) the area along the Yellow, Wei, and Fen Rivers in the Guanzhong Plain and southwestern Shanxi; and

(3) the Yangtze River delta, which was the most economically advanced area in the late Ming, contributing over 17% to the

nation's total tax revenue (Liang, 2008). These areas belonged to North China, Northwest China, and the Yangtze-Huai

Region, respectively.

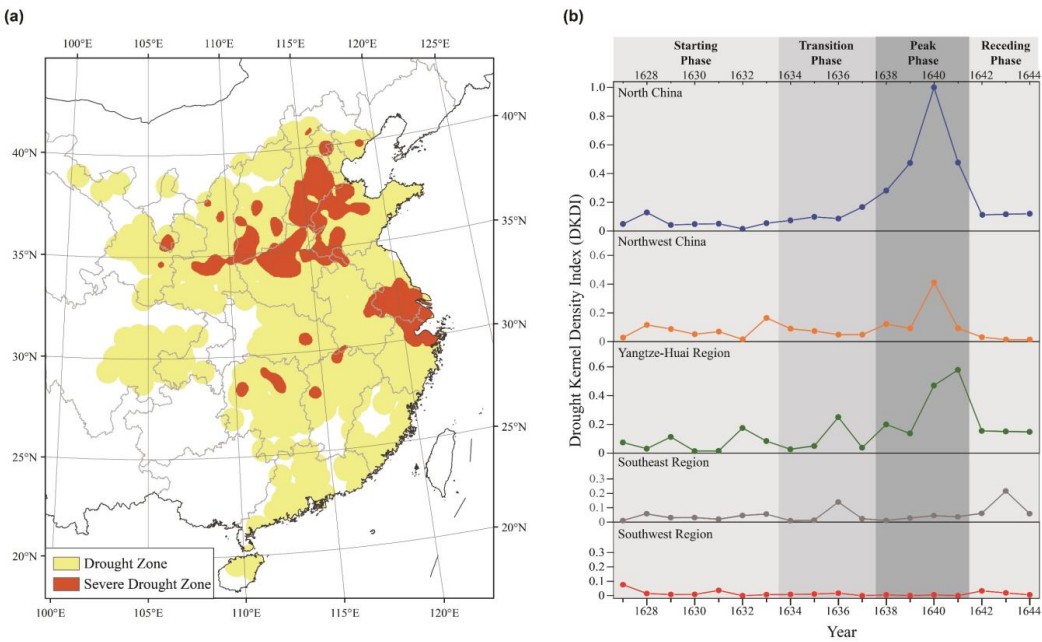

**Figure 2: Spatial and Temporal Patterns of Drought, 1627-1644**

**(a) Overall spatial distribution of drought, 1627-1644. (b) Drought Kernel Density Index (DKDI) series for five regions**



Figure 2(b) presents the DKDI series for the five subregions within the study area. The data reveal that North China, Northwest China, and the Yangtze-Huai Region were primarily affected by the drought. In contrast, the DKDI values for the Southeast and Southwest Regions were comparatively low, suggesting these areas experienced less severe drought conditions.

Notably, the progression of drought exhibits significant regional variations, indicating diverse evolutionary patterns across the different regions. The DKDI series for North China resembles a single-peaked curve. Between 1633 and 1638, the DKDI gradually increased each year, although at a slow pace; after 1638, there was a marked surge. The peak of this series appeared in 1640, a year characterized by both the maximum spatial extent and the highest intensity of the drought. From 1638 to 1641, drought affected the entire region almost annually, with a notable decline after 1642. During the early

Chongzhen period, Northwest China, particularly Shaanxi Province, faced severe and prolonged drought until 1634. The DKDI gradually decreased afterward but surged again after 1638, also peaking in 1640. The DKDI series for the Yangtze-Huai Region depicted a multi-peaked curve, reflecting continuous cycles of drought aggravation and alleviation. Before 1638, significant fluctuations occurred, with peaks in 1629, 1632, and 1636. Each drought event lasted one year, followed by relief. After 1638, the DKDI rose again, and the drought lasted for a long time. The DKDI reached its top in 1641, a year

later than North and Northwest China.

Overall, the Chongzhen Drought is divided into four phases by analyzing its spatial-temporal evolution (Fig. 2b):

(1) Starting Phase (1627-1633): The primary drought zone was situated in Northwest China, particularly in Shaanxi and Shanxi provinces.

(2) Transition Phase (1634-1637): There was a gradual shift in the main drought zones from Northwest China to North

China.

(3) Peak Phase (1638-1641): The drought zones in both the north and south became interconnected, extending across all areas north of 29°N.

(4) Receding Phase (1642-1644): The drought contracted in scope and weakened in intensity.

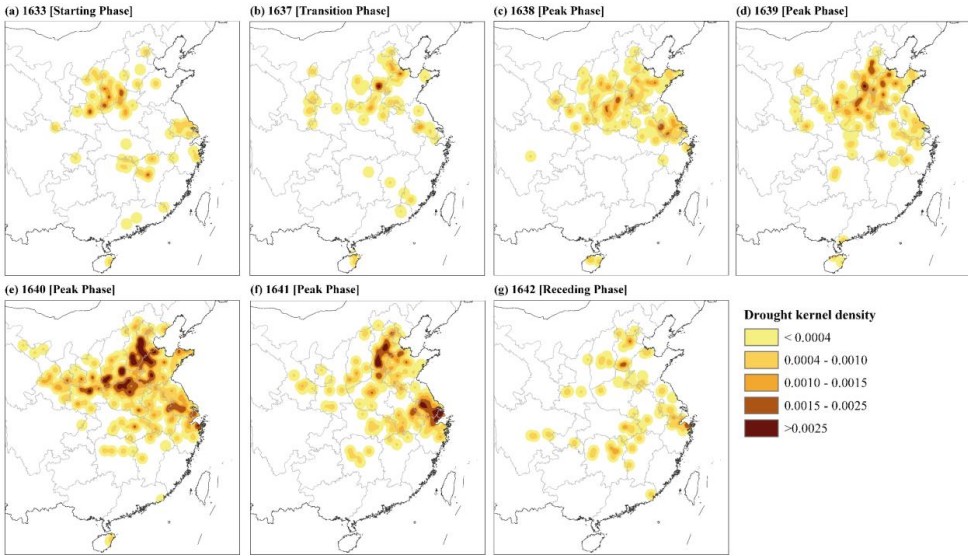

**Figure 3: Spatial distribution of drought during representative years of each phase**

We also reconstructed the spatial distribution pattern of drought at annual resolution (see Appendix B for details). Figure 3 illustrates these patterns across select years, highlighting the drought's progression through its distinct phases. In the





starting phase, the drought was concentrated in the northern part, particularly around Shaanxi Province in Northwest China, where it was extensive, intense, and prolonged. During the transition phase, the drought in Northwest China eased, while in North China, especially in Henan and Hebei provinces, the drought's extent grew annually. During the peak phase (1638-1641), the drought area in the north merged with that in the Yangtze River basin, continuously expanding. Meanwhile, the drought intensity markedly increased, with a greater proportion of Level 3 and 4 drought events. The year 1640 witnessed the broadest and most intense drought conditions, with 367 counties reporting droughts, 69% of which in Level 3 or 4. The drought extended to Changsha in the south, the coast in the east, Beijing in the north, and the Hexi Corridor in the west. Significant water bodies dried up, including sections of the Yellow River in Henan and Jiangsu, the Fen and Zhang Rivers in Shanxi, and the Wen and Si Rivers in Shandong, as well as Baiyangdian, North China's largest freshwater lake. In 1641, the drought slightly eased in the north but intensified in the south. In the Yangtze River Delta, records indicated more than four months without rain during the summer and autumn—highly unusual for the area's climatic norms. Rivers such as the Suzhou Creek dried up. The record "no water even when drilling a well at the bottom of the river" indicated that the groundwater levels had dropped significantly. The receding phase began in 1642, with a notable reduction in drought severity and a contraction of the affected areas. Overall, from 1627 to 1644, the primary zone of extreme drought shifted from Northwest China to North China and then expanded southwards.

It is important to note that the drought zones identified in this study were reconstructed using historical document records. However, the northern and northwestern parts of the study area were near the borders of the Ming Dynasty at that time, as illustrated in Figure 1. Conflicts are frequent here during the Chongzhen era, contributing to social instability. The unrest resulted in a significant scarcity of historical documentation in these areas. Consequently, the actual extent of the drought may have been more widespread towards the west and north than the records suggest.

## 4 The evolution of famine and comparison with drought

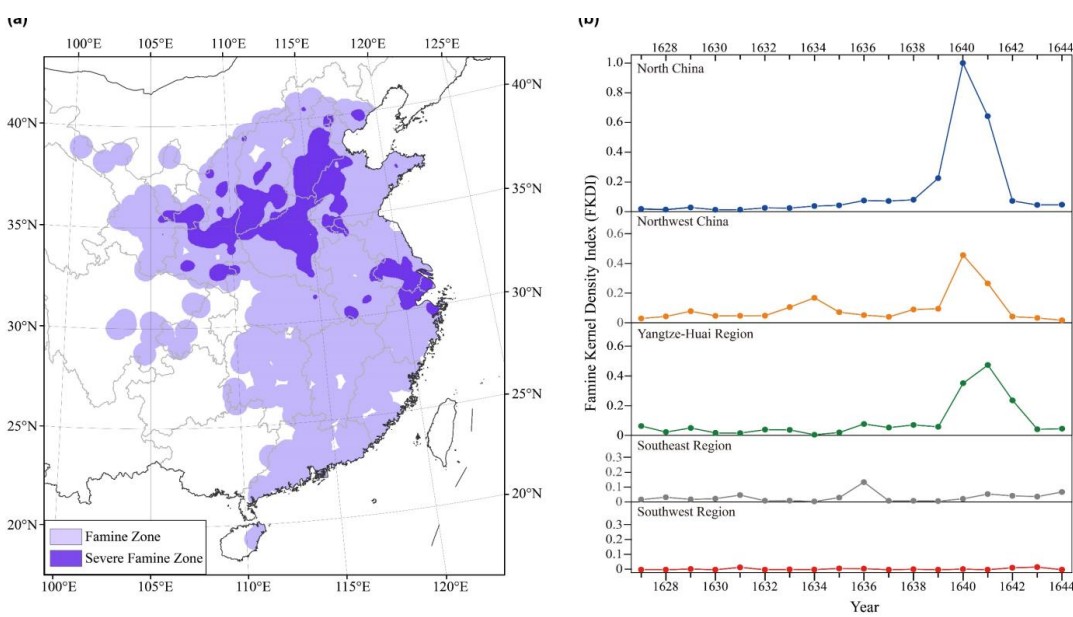

**Figure 4: Spatial and Temporal Patterns of Famine, 1627-1644**

**(a) Overall spatial distribution of famine, 1627-1644. (b) Famine Kernel Density Index (DKDI) series for five regions**



Figure 4(a) presents the general spatial distribution of famine during 1627-1644, indicating that famine affected most areas of central and eastern China. The extent of the famine zone closely matched that of the drought zone. All severe famine zones located north of 29°N, largely overlapped with the three areas experiencing severe drought.

Figure 4(b) presents the Famine Kernel Density Index (FKDI) series for five regions from 1627 to 1644. North China, Northwest China, and the Yangtze-Huai Region were significantly affected by famine, exhibiting higher FKDI values. In contrast, the Southeast and Southwest Regions experienced milder famine conditions, with FKDI peaks not exceeding 0.2. A comparison of the FKDI and DKDI series reveals a generally consistent trend in the development of famine and drought across the regions. In the starting phase of the drought, FKDI was comparatively high in Northwest China. During the transition phase, FKDI in North China grew, but at a slow pace. The peak phase saw the FKDI reaching its maximum in North China, Northwest China, and the Yangtze-Huai Region in the years 1640, 1640, and 1641, respectively, coinciding with the peak years of DKDI in these regions. In the receding phase, FKDI showed a downward trend as the drought eased.

Although the trends exhibit similarities, it is important to recognize that the FKDI and DKDI series differ significantly in two key aspects.

Firstly, the FKDI series is smoother than the DKDI series, characterized by fewer fluctuations. Specifically, several peaks observed in the DKDI series before 1640 are absent in the FKDI series. For example, the DKDI series for North China peaks in 1628, and the DKDI for the Yangtze-Huai Region peaks in 1629, 1632, and 1636, yet these peaks have no equivalents in the FKDI series. This discrepancy suggests that not all drought events escalate into famines.

Secondly, while the highest peaks of the FKDI and DKDI series occurred in the same years for Northwest China, North China, and the Yangtze-Huai Region (1640, 1640, and 1641, respectively), the FKDI peaks were broader and exhibited a right-skew. It means that although both DKDI and FKDI values decreased in the year following their peak, the decline in FKDI was less pronounced than that of the DKDI. Despite the reduction in drought intensity, society was unable to promptly recover from its devastated state, resulting in the continued prevalence of famine. This pattern indicates a continuity in the impact of drought on famine.

## 5 The statistical relationship between drought and famine

To examine if there is a statistically significant relationship between drought and famine, a Spearman correlation analysis was conducted on the DKDI and FKDI data from Northwest China, North China, and the Yangtze-Huai Region. The results, presented in Table 3, reveal a significant positive correlation between DKDI and FKDI, with a correlation coefficient of 0.710. This indicates that the severity of drought is closely associated with the likelihood of severe famine. Additionally, the DKDI and FKDI of the subsequent year also exhibit a significant positive correlation, though the correlation coefficient decreases to 0.583. However, the correlation between DKDI and FKDI in the third year is not significant. These results underscore the ongoing impact of drought on famine, lasting until the following year, albeit at a reduced intensity.

**Table 3: The correlation analysis results between DKDI and FKDI at the regional scale**

|  | $FKDI_{(t)}$ | $FKDI_{(t+1)}$ | $FKDI_{(t+2)}$ |
|---|---|---|---|
| $DKDI_{(t)}$ | 0.710[**] | 0.583[**] | 0.268 |

[**] indicates significant correlation (two-tailed, $p < 0.01$)

To further investigate the potential regional variations in the relationship between drought and famine, North China, Northwest China, and the Yangtze-Huai Region were subdivided into 16, 14, and 11 sub-provincial zones, respectively, for downscaling studies. This subdivision considered both physical and human geographical features, referencing China's provincial regionalization schemes (detailed in Appendix C). The methodology described in Sect.2.4 was employed to calculate the DKDI and FKDI for each sub-provincial zone, followed by Spearman correlation analysis (Table 4). At the sub-



province scale, a significant positive correlation exists between DKDI and FKDI for the current year, the following year, and the third year, with a 99.9% confidence level. However, the correlation coefficients show a year-to-year decline. This pattern suggests a continuity of drought's impacts on famine at the sub-provincial scale, lasting up to the third year, but the magnitude decays over time. This finding holds for all three major regions of Chongzhen Drought.

**Table 4: The correlation analysis results between DKDI and FKDI at the sub-provincial scale**

|  |  | $FKDI_{(t)}$ | $FKDI_{(t+1)}$ | $FKDI_{(t+2)}$ |
|---|---|---|---|---|
| North China |  | $0.626^{**}$ | $0.559^{**}$ | $0.434^{**}$ |
| Northwest China | $DKDI_{(t)}$ | $0.746^{**}$ | $0.641^{**}$ | $0.412^{**}$ |
| Yangtze-Huai Region |  | $0.648^{**}$ | $0.529^{**}$ | $0.310^{**}$ |

$^{**}$ indicates significant correlation (two-tailed, $p<0.01$)

At the sub-provincial scale, regression analysis was carried out using DKDI as the independent variable and FKDI as the dependent variable. Since both scatter plots and correlation analysis showed a linear relationship between the two variables, a unary linear regression model was selected. The obtained regression equation is presented below:

$$FKDI = 0.668DKDI + 0.001$$

The regression model demonstrates statistical significance ($p < 0.001$), with a coefficient of determination ($R^2$) of 0.673. This indicates that approximately 67.3% of the variation in the FKDI series can be explained by changes in the DKDI. Consequently, drought is the primary factor influencing famine, accounting for roughly two-thirds of its variability.

Table 5 presents the results of regression analyses across different regions. No matter which region, the regression equations achieved statistical significance at the 99.9% level, all with positive regression coefficients. Based on the

coefficient of determination ($R^2$), the contribution of DKDI to FKDI in North China, Northwest China, and the Yangtze-Huai Region is 73.4%, 69.3%, and 63.3%, respectively. Among the three principal disaster-affected regions, drought had the highest impact on famine in North China.

**Table 5: The regression analysis results of DKDI and FKDI at the sub-provincial scale**

| Region | Regression equation | $R^2$ | P-value |
|---|---|---|---|
| North China | $FKDI = 0.764DKDI - 0.007$ | 0.734 | <0.001 |
| Northwest China | $FKDI = 0.822DKDI + 0.006$ | 0.693 | <0.001 |
| Yangtze-Huai Region | $FKDI = 0.446DKDI + 0.005$ | 0.633 | <0.001 |

## 6 Discussion

### 6.1 The transmission process of extreme drought impacts

In ancient China, agriculture was pivotal to the nation's foundation, and also a critical link in the transmission of climate change impacts to human society. The severity of drought's impact on agriculture largely depends on the season it occurs. In this study, 886 drought records provided specific seasonal or monthly details. This facilitated an analysis of the seasonal distribution of droughts in North China, Northwest China, and the Yangtze-Huai Region from 1627 to 1644 (Figure

5). In North China, summer experienced the highest frequency of droughts, constituting 73% of the total occurrences, followed by spring and autumn with 34% and 23%, respectively. Consecutive droughts spanning spring to summer or summer to autumn were also common. The region primarily cultivates two types of food crops: cereals like millet and sorghum, planted in spring and harvested in autumn; and winter wheat, sown in autumn and harvested in the next summer (Han, 2000). Summer serves as the critical growth period for cereals, whereas autumn and spring cover the tillering and

grain-filling stages of winter wheat with huge water requirements (Zhong et al., 2000). That is to say, most drought events





coincided with the crucial crop growth periods in North China, resulting in inadequate water supply for crops. A similar situation is observed in the other regions. The most drought-prone seasons in Northwest China are summer (62%) and autumn (36%), which meet the primary growth periods for spring wheat. In the Yantze-Huai Region, the majority of droughts occurred in summer (73%) and autumn (41%). The main food crop in the region is rice, which is cultivated in

spring, transplanted in summer, and harvested in autumn, with a huge water demand. Droughts during these growth seasons often lead to diminished crop yields or complete failures, subsequently reducing the per capita food availability. Famine ensues when the food holding of most people fails to meet their basic needs for survival and health. Hence, the dominant pathway through which climate change impacts translate into social consequences is as follows: extreme drought → declining agricultural harvest → food shortage → famine.

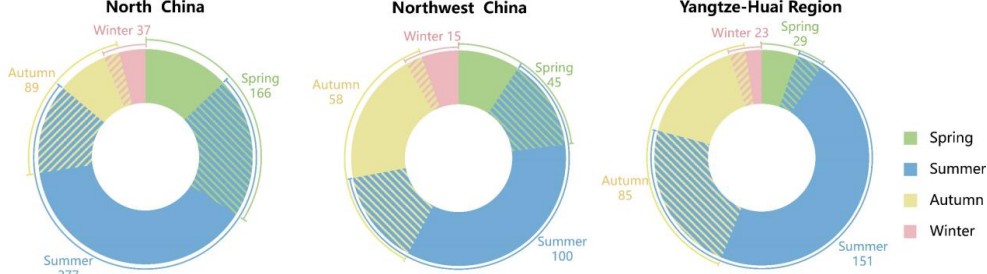


**Figure 5: Seasonal distribution of drought records**

Series and correlation analyses reveal that droughts have continuous impacts on famine, although these effects diminish over time. The reason is that the impacts of drought, once transmitted to societal levels, disrupt the production, population, and economic systems to various extents. Even as the drought eased and precipitation normalized, the society

remained damaged, hindering the immediate restoration of production and life order. Historical records underscore this phenomenon. For instance, in 1641, there was a record that "at that time, there was land, but no people and cattle, so the land was out of cultivation and turned back barren" in Neihuang, Henan Province. As many people died and cattle were sold or eaten, there was a shortage of both labor and production means, so it was hard to resume agricultural activities. Similarly, a record from Xinzheng said: "In spring, famine followed by plague, the dead seven out of ten. In May, although the wheat is

ripe, no one to harvest". These examples illustrate that the societal devastation led by drought is a persistent process. When drought is over, it takes some time for the society to return to normalcy from disaster damage.

The comparison of the DKDI and FKDI series indicates that not all drought events turn into famine. It is particularly evident in the Yangtze-Huai Region, where the DKDI series identifies peaks in 1629, 1632, 1636, and 1638, but the FKDI series does not present corresponding peaks. This discrepancy suggests that although extreme droughts occurred in the

Yangtze-Huai Region in these years, their impacts decreased or even dissipated before reaching the famine level, largely due to human societies being active in developing various coping measures (Yang et al., 2021). Between 1627 and 1644, the Yangtze-Huai Region documented at least 15 categories of response measures (Table 6), the most varied and frequent across all regions studied. The actors of measures (1)~(6) were individual people. Measures (1)~(2) operated in the production subsystem, aiming to reduce the impacts of droughts and locusts on food production. The objective of measures (3)~(6) was

to enhance food accessibility, to cope with food shortages caused by crop failures. On the other hand, measures (7)~(15) were government-driven. Measures (7)~(9) distributed stored grain to the hungry in different ways, while measures (10)~(11) left a greater proportion of grain to producers through tax reductions, all of which were essentially aimed at increasing the per capita share of food of local people. Additionally, measures (12)~(14) dealt with the social influences of famine, including plague, displacement, and population mortality, while measure (15) could calm down the public to some

degree, so as to maintain a stable social order. These measures worked on different segments of the socio-ecological system,





effectively mitigating the impacts of the droughts and preventing, or at least curtailing, their spread to higher-level subsystems, demonstrating the Yangtze-Huai Region's resilience to the drought impacts. While all regions have certainly tried to make use of all their capacities and possible measures, the limits of social resilience were challenged when the drought impacts exceeded the system's threshold (Xu et al., 2021). From 1640 to 1641, 110 counties within the Yangtze-Huai Region reported drought, with Level 3 and 4 drought events accounting for more than 65%. The Yangtze River Delta even recorded "little rain for the whole year". While the above response measures still exist, they were only a drop in the bucket for local people. Severe famine swept through the region, and there were even instances of cannibalism.

**Table 6: Response measures in the Yangtze-Huai Region from 1627 to 1644**

|  | Response Measure | Main Actor | Meaning |
|---|---|---|---|
| (1) | Reseeding | Farmer | After the crop dies due to drought, re-plant some if rainfall occurs |
| (2) | Locust catching | Farmer | Catch locusts to prevent them from destroying crops. Sometimes local government also encouraged people to do so and gave them some grain or money as an award. |
| (3) | Food substitution | Famine victim | Eat wild herbs, chaff, grass roots, tree bark, soil, and so on to satisfy their hunger. The most extreme case is cannibalism (i.e. some people kill others and eat corpses). |
| (4) | Selling property | Famine victim | Selling property such as houses, land, cattle, agricultural tools, etc., in exchange for money to buy food. The most extreme case is trafficking women and children. |
| (5) | Displacement | Famine victim | Leave their hometown and flee to surrounding areas in search of food |
| (6) | Robbery | Famine victim | Rob on the road, or rob the homes of the wealthy, landlords, and gentries to obtain food or money. |
| (7) | Giving porridge | Local government, officials, gentries | Open porridge factories to feed famine victims |
| (8) | Selling grain at low price | Local government | Sell stored grain at low prices to prevent excessive increases in grain prices on the market due to shortages. |
| (9) | Financial or food relief | Local government, central government | Distribute food or money to the victims directly. |
| (10) | Tax exemption | Central government | A discretionary exemption from taxes or corvees. |
| (11) | Tax substitution | Central government | Allow disaster areas to convert the tax grain into money or other items to hand in. |
| (12) | Treatment | Doctors, local government | Distribute medicine to the population if a plague occurred, or hire doctors to treat the patients. |
| (13) | Corpse management | Local government, officials, gentries | Distribute coffins to the families of the deceased, or bury the corpses together. |
| (14) | Hosting of orphans | Local government | Set up shelters to take in abandoned kids. |
| (15) | Praying for rain | Local officials | Offer sacrifices to gods and pray for rain. |

## 6.2 Other factors triggering and exacerbating famine

Regression analysis indicates that drought was the primary cause of famine in this case, contributing about two-thirds. However, it was not the only cause; other natural and social factors also played significant roles. (1) On a long-time scale, the late Ming was a period of climate cooling in the Little Ice Age. Between the 1550s and 1650s, the mean temperature of China decreased by about 0.4°C/100 yr; the mean temperature in North China was about 0.7°C lower than the 1951-2000 average (Ge et al., 2013). This cooling shortened the growing season for crops and reduced arable land in the north. As a result, there had been an overall decline in per capita food availability, making society more sensitive to external forcings, including climate extremes (Fang et al., 2014). (2) Other disasters, such as localized floods, windstorms, frosts, and large-scale locust infestations in the context of persistent droughts (Fig.6a). These disasters, compounded with droughts, exacerbated the damage to agricultural production. Against the backdrop of massive famine, there was a severe plague during the Chongzhen period (Fig. 6b), which led to numerous deaths and made the resumption of social order more difficult





(Cao, 1997). (3) The Ming government's declining governance capacity evidenced both politically and economically. Political issues included the emperor's negligence, abuse of power by eunuchs, power struggles among central officials, and corruption among local officials. Economic problems included revenue shortages, overspending, reduced warehousing, and increased tax rates (Bai, 2004; Twitchett and Fairbank, 1998). These issues limited the government's ability to respond to

large-scale droughts effectively. (4) The growing border crisis. From the 16th century onwards, wars between the Mongols and the Ming Dynasty increased significantly (Fu et al., 1986). At the same time, the Manchu in the northeast gradually developed under the leadership of Nurhaci, and in 1618 declared war on the Ming Dynasty. The prolonged war strained financial resources and diverted funds from disaster relief. It's notable that climate change also played a role in the border crisis. Climate cooling led to grassland degeneration and desertification in northern China, reducing the production capacity

of the Mongols and Manchus and pushing them to invade the Ming Dynasty's territory for resources (Ge, 2011). These factors, in conjunction with the primary impact of drought, illustrate a complex web of causes behind the famine, highlighting the interplay between natural events and human society.

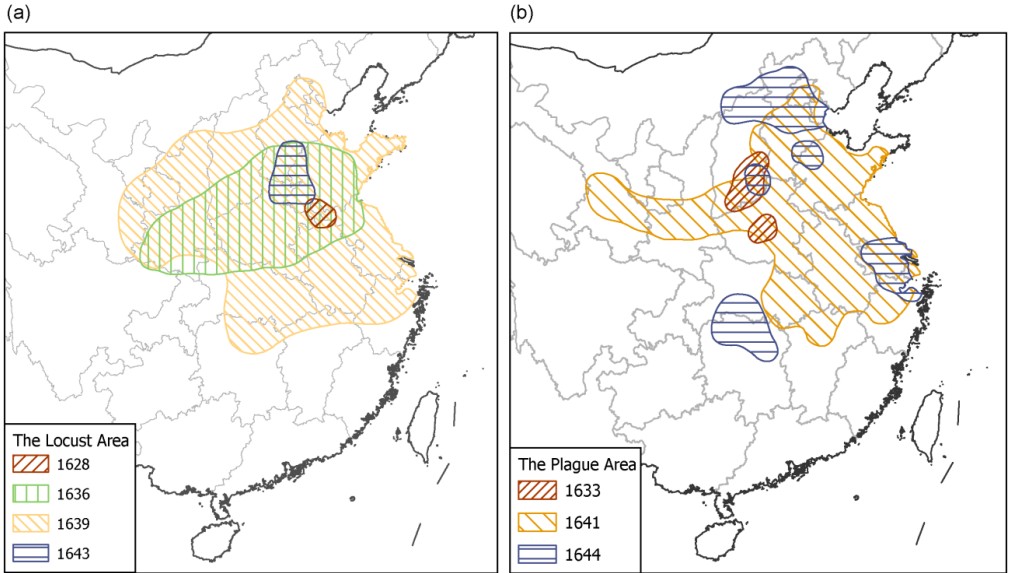

**Figure 6: Areas affected by the locust infestation and plague**

**(a) Locust infestation (b)Plague**

A coupled socio-ecological system (Gallopin, 2006) comprises three tiers: natural system, support system, and human system (Bossel,1999; Fang et al., 2014). Typically, the effects of climate change propagate through these levels sequentially. Yet, this transmission does not follow a simple cause-and-effect pattern. Instead, climate change acts as an external force on the socio-ecological system, interacting with the vulnerability of society and triggering a cascade of feedback mechanisms.

In the case of Chongzhen Drought, the core event in the natural system was the persistent extreme droughts; the core event in the support system was the decline in food production; and the initial core event in the human system was famine. Famine further triggered a sequence of social consequences, including peasantry uprisings, plagues, widespread mortality, decreased social stability, and ultimately contributed to the collapse of the Ming Dynasty (Zheng et al., 2014; Tan et al., 2011b). As can be seen, when the effects of climate change ascend to higher levels, the outcomes tend to be more complex and regulated by

more non-climatic factors (e.g. economy, policy).


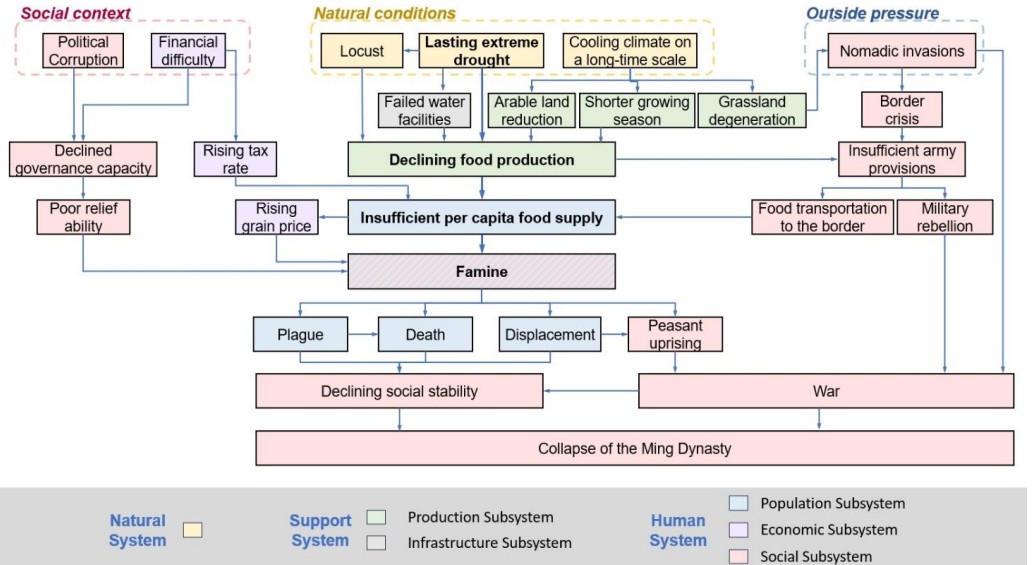

**Figure 7: The cascading transmission paths of the impact of the Chongzhen Drought**

## 7 Conclusion

Based on historical records and employing kernel density estimation, this study has reconstructed the spatial
distribution patterns of drought from 1627 to 1644, alongside developing the Drought Kernel Density Index (DKDI) series
for five regions in China. This extreme drought event lasted for 18 years and affected most of central and eastern China
spatially. Drought is relatively severe in North China, Northwest China, and the Yangtze-Huai Region, mainly located north
of 29°N. The analysis reveals notable regional differences in drought evolution. In North China, the DKDI followed a single-
peaked trajectory, with drought intensifying from 1633 and reaching its peak in 1640. Northwest China experienced the
earliest onset of drought, which eased in the middle period before peaking again in 1640. The Yangtze-Huai Region
displayed a multi-peaked DKDI, undergoing several cycles of worsening and easing drought, with the most severe drought
occurring in 1641, a year later than in the northern regions. The temporal and spatial progression of the Chongzhen Drought
can be divided into four phases: the starting phase (1627-1633), where the main drought zone was in Northwest China; the
transition phase (1634-1637), during which the drought zone gradually shifted to North China; the peak phase (1638-1641),
where drought zones in the north and south converged and expanded to all areas north of 29°N; and the receding phase
(1642-1644), marked by a weakening of the drought. The year 1640 witnessed the broadest and most intense drought
conditions, during which numerous significant water bodies, including sections of the Yellow River, dried up. From 1627 to
1644, there was a shift of the primary drought zones from Northwest China to North China, and then expanding southwards.
Summer and autumn, the critical crop-growing seasons in the study area, experienced the highest frequencies of drought,
significantly impacting food production.

In the case of the Chongzhen Drought, drought was the primary factor in triggering famine, with the spatial and
temporal patterns of both phenomena displaying remarkable similarities. Spatially, the area affected by famine from 1627 to
1644 significantly overlapped with that experiencing drought. Temporally, the progression of drought and famine exhibited a
general consistent. The years when the Famine Kernel Density Index (FKDI) reached its peak in North China, Northwest
China, and the Yangtze-Huai Region coincided with the peaks of the DKDI series. Moreover, a significant positive



correlation between DKDI and FKDI was observed. Regression analysis revealed that, on a sub-provincial scale, drought accounted for approximately 67.3% of the famine's causation, increasing to as much as 73.4% in North China. Series and correlation analyses further demonstrated the continuous impacts of drought on famine, which could persist for 2-3 years.

Extreme drought → declining agricultural harvest → food shortage → famine is the dominant transmission pathway in which the climate change impacts reached the human system in this case. However, the transmission of climate change impacts is characterized by a non-linear process with multiple feedback mechanisms. In addition to extreme drought, other natural factors and socio-economic contexts also play significant roles in the occurrence and development of famines. Moreover, human responses regulated the transmission of drought impacts. In the Yangtze-Huai Region, particularly, multiple response measures curtailed the spread of drought impacts to the human system before 1640. This thorough study

deepens our understanding of historical extreme climate events and their social impacts, while also underscoring the significance of grasping the interaction between natural dynamics and human actions when dealing with climate challenges.

**Data Availability**

All the data used in this study are described in Appendix A.

**Author Contribution**

SC and YS contributed to the study conception and methodology design. SC collected data, performed most of the analysis, and wrote the initial draft. YS performed the funding acquisition and supervised the whole study. XC drafted the writing outline and contributed to the data visualization. LEY proposed ideas for the discussion section and revised the manuscript. All authors read and approved the final manuscript.

**Competing Interests**

The authors declare that they have no conflict of interest.

**Acknowledgements**

We would like to thank all colleagues in our research group from Beijing Normal University in China and Ludwig Maximilian University of Munich (LMU) in Germany. We would like to thank Yixin Jin for her help with the language. We would like to thank the reviewers and editors for their comments.

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





**Appendix A. Sources of historical records used in this study**

| No. | Title of the source | Provinces involved | Author and time | The number of records |
|---|---|---|---|---|
| 1 | A compendium of Chinese meteorological records of the last 3000 years (中国三千年气象记录总集) | The whole study area | Zhang De'er (张德二), 2004 | 5006 |
| 2 | Historical materials of drought and flood in North China and Northeast China for the past 500 years (华北、东北近五百年旱涝史料) | Beijing, Tianjin, Hebei, Shandong, Henan, Shanxi | Chinese Academy of Meteorological Sciences (中央气象局研究所) et al., 1975 | 127 |
| 3 | Climatic historical materials for the last 500 years in East China (华东地区近五百年气候历史资料) | Shanghai, Jiangsu, Anhui, Zhejiang, Jiangxi, Fujian | Shanghai Meteorological Administration (上海市气象局) et al., 1978 | 201 |
| 4 | Climatic historical materials of Hubei province for the last 500 years (湖北省近五百年气候历史资料) | Hubei | Wuhan Regional Climate Center (武汉区域气候中心), 2018 | 55 |
| 5 | Climatic historical data of Jiangxi province (江西省气候史料) | Jiangxi | Reference room of Jiangxi provincial meteorological administration (江西省气象局资料室), 1978 | 45 |
| 6 | Historical materials of drought and flood in Sichuan province for the past 500 years (四川省近五百年旱涝史料) | Sichuan, Chongqing | Reference room of Sichuan Provincial Meteorological Administration (四川省气象局资料室), 1978 | 29 |
| 7 | Chronology of large floods and droughts in Henan Province in past dynasties (河南省历代大水大旱年表) | Henan | Henan provincial hydrologic general station (河南省水文总站), 1982 | 174 |
| 8 | Hydrologic and climatic data of Henan Province in past dynasties: including drought, flood, locust, wind, hail, frost, snow, cold, heat (河南省历代旱涝等水文气候资料：包括旱、涝、蝗、风、雹、霜、大雪、寒、暑) | Henan | Henan provincial hydrologic general station (河南省水文总站), 1982 | 25 |
| 9 | Historical famine and disaster records in Beijing: from 80 BC to 1948 AD (北京历史灾荒灾害纪年：公元前80年-公元1948年) | Beijing | Yu Deyuan (于德源), 2004 | 15 |
| 10 | History of Natural disasters in Shanghai (上海自然灾害史) | Shanghai | Liu Changsen (刘昌森) et al., 2010 | 41 |
| 11 | Historical materials of natural disasters in Shaanxi Province (陕西省自然灾害史料) | Shaanxi | Meteorological Station of the Shaanxi Provincial Meteorological Administration (陕西省气象局气象台), 1976 | 21 |
| 12 | A brief chronicle of disasters in Zhejiang. Zhejiang. Zhejiang People's | Zhejiang | Chen Qiaoyi (陈桥驿), 1991 | 21 |





| | | | | |
|---|---|---|---|---|
| | Publishing House (浙江灾异简志) | | | |
| 13 | History of Disasters in Shanxi (山西灾害史) | Shanxi | Wang Jianhua (王建华), 2014 | 66 |
| 14 | Chronology of natural disasters in Hunan (湖南自然灾害年表) | Hunan | Hunan provincial institute of cultural relics and archaeology (湖南历史考古研究所), 1961 | 25 |
| 15 | History of natural disasters in Shandong province (山东省自然灾害史) | Shandong | Wei Guangxing (魏光兴), 2000 | 30 |
| 16 | Historical materials of natural disasters in Guangdong Province, the revised and enlarged edition (广东省自然灾害史料·增订本) | Guangdong, Hainan | Guangdong research institute of culture and history (广东省文史研究馆), 1963 | 27 |
| 17 | The literature and history data of Gansu Province (甘肃文史资料) | Gansu | Information Research Committee of Gansu Provincial Committee of the Chinese People's Political Consultative Conference (中国人民政治协商甘肃省委员会资料研究委员会), 1985 | 33 |
| 18 | The history of disasters and famines in Northwest China (西北灾荒史) | Shaanxi, Gansu, Ningxia | Yuan Lin (袁林), 1994 | 86 |
| 19 | Historical materials of natural disasters in Haihe River basin (海河流域历代自然灾害史料) | Beijing, Tianjin, Hebei | Hebei provincial drought and flood forecasting team (河北省旱涝预报课题组), 1985 | 117 |
| 20 | Chinese Meteorological Disasters Ceremony: Shandong Volume (中国气象灾害大典·山东卷) | Shandong | Wen Kegang (温克刚) et al., 2006 | 60 |
| 21 | Chinese Meteorological Disasters Ceremony: Hunan Volume (中国气象灾害大典·湖南卷) | Hunan | Wen Kegang (温克刚) et al., 2006 | 35 |
| 22 | Chinese Meteorological Disasters Ceremony: Gansu Volume (中国气象灾害大典·甘肃卷) | Gansu | Wen Kegang (温克刚) et al., 2005 | 23 |
| 23 | Chinese Meteorological Disasters Ceremony: Ningxia Volume (中国气象灾害大典·宁夏卷) | Ningxia | Wen Kegang (温克刚) et al., 2007 | 15 |
| 24 | Chinese Meteorological Disasters Ceremony: Hainan Volume (中国气象灾害大典·海南卷) | Hainan | Wen Kegang (温克刚) et al., 2008 | 5 |





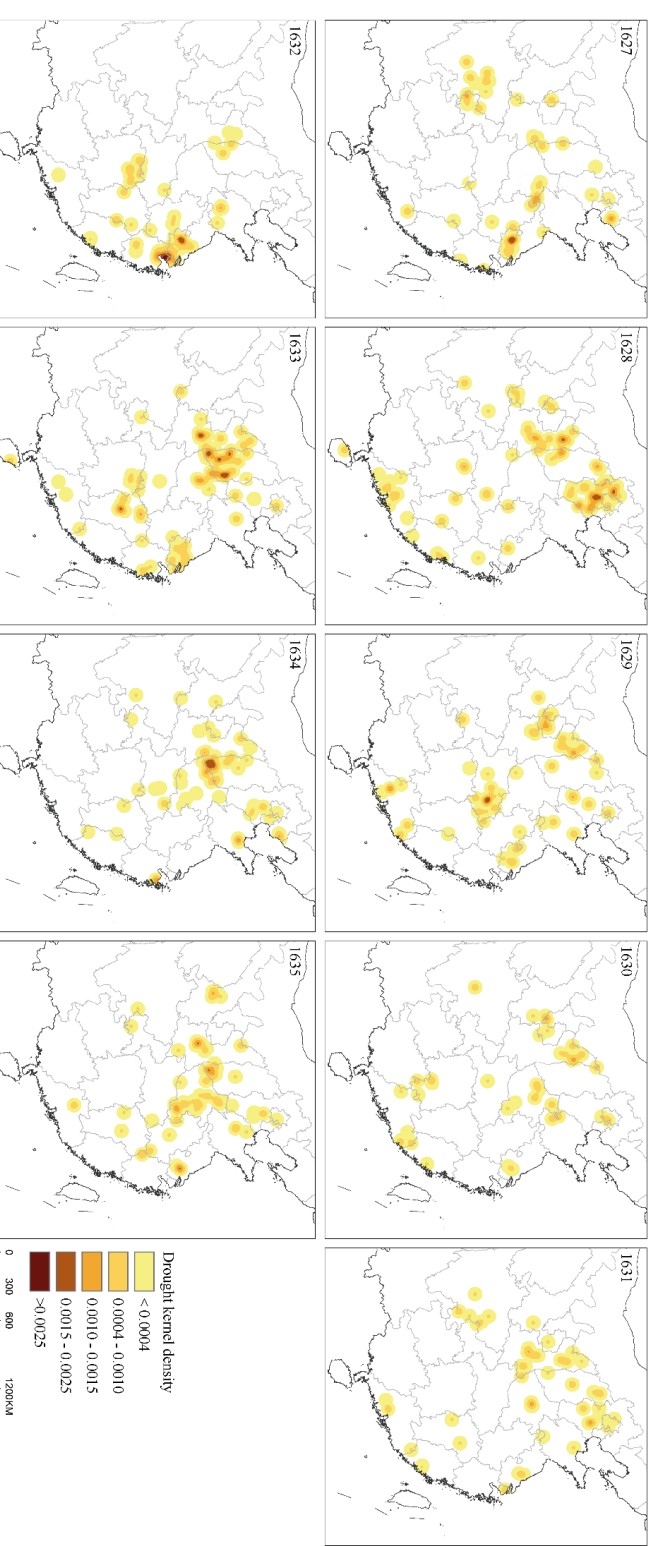

**Appendix B. Year-by-year spatial distribution of drought during 1627-1644**



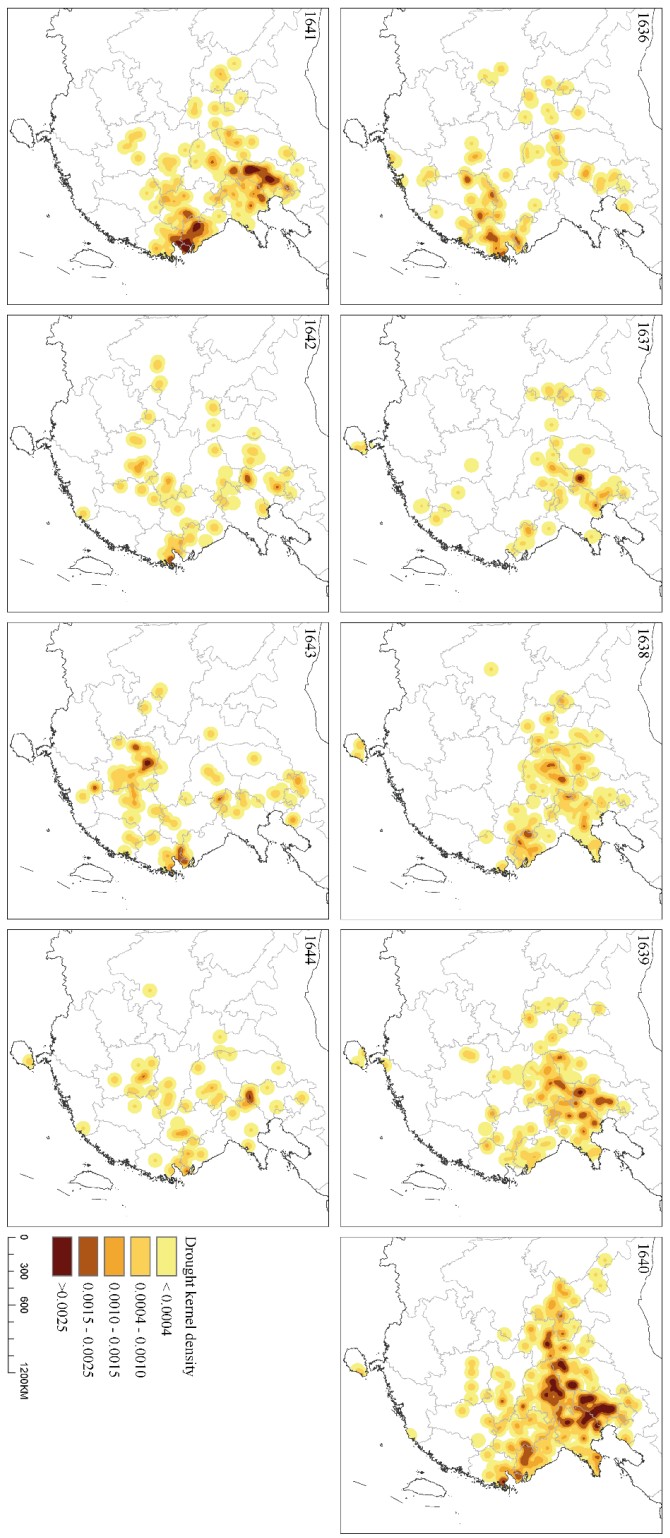





**Appendix C. The division of sub-provincial zones**

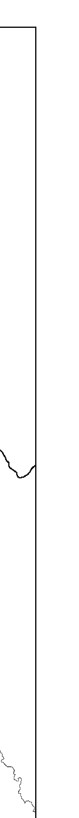

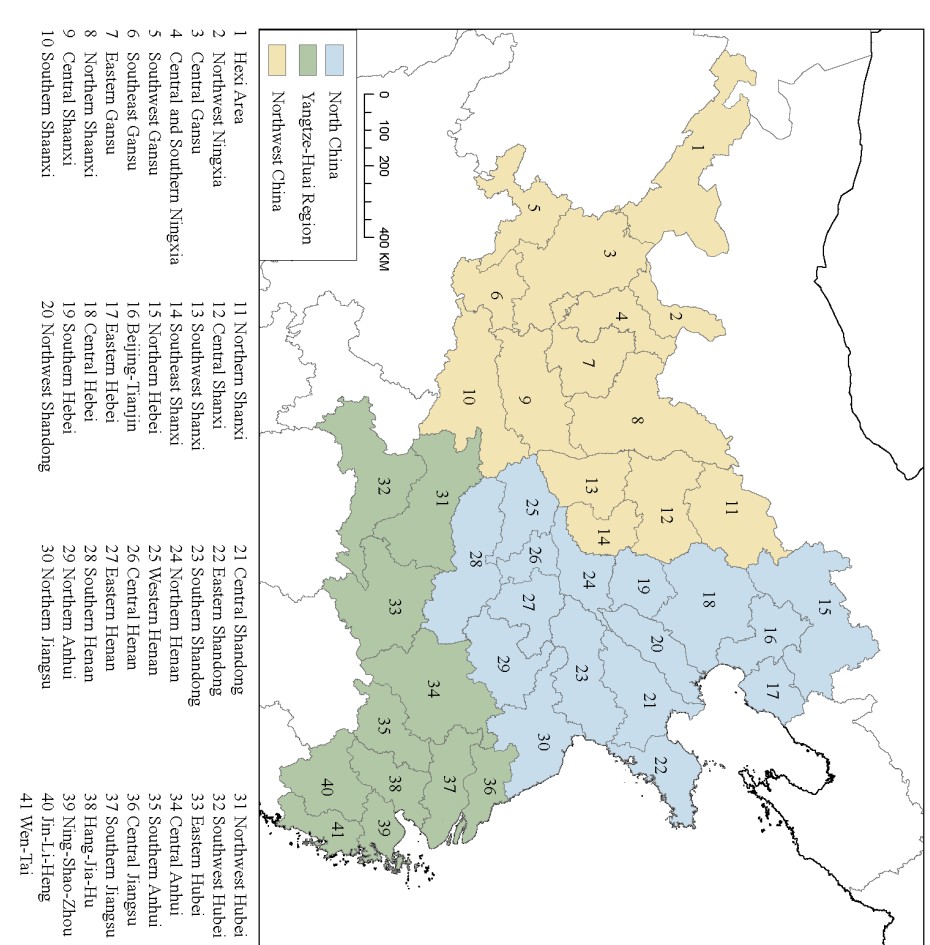

Hexi Area
Northwest Ningxia
Central Gansu
Central and Southern Ningxia
Southwest Gansu
Southeast Gansu
Eastern Gansu
Northern Shaanxi
Central Shaanxi
Southern Shaanxi

Northern Shanxi
Central Shanxi
Southwest Shanxi
Southeast Shanxi
Northern Hebei
Beijing-Tianjin
Eastern Hebei
Central Hebei
Southern Hebei
Northwest Shandong

Central Shandong
Eastern Shandong
Southern Shandong
Northern Henan
Western Henan
Central Henan
Eastern Henan
Southern Henan
Northern Anhui
Northern Jiangsu

Northwest Hubei
Southwest Hubei
Eastern Hubei
Central Anhui
Southern Anhui
Central Jiangsu
Southern Jiangsu
Hang-Jia-Hu
Ning-Shao-Zhou
Jin-Li-Heng
Wen-Tai

Northwest China
North China
Yangtze–Huai Region



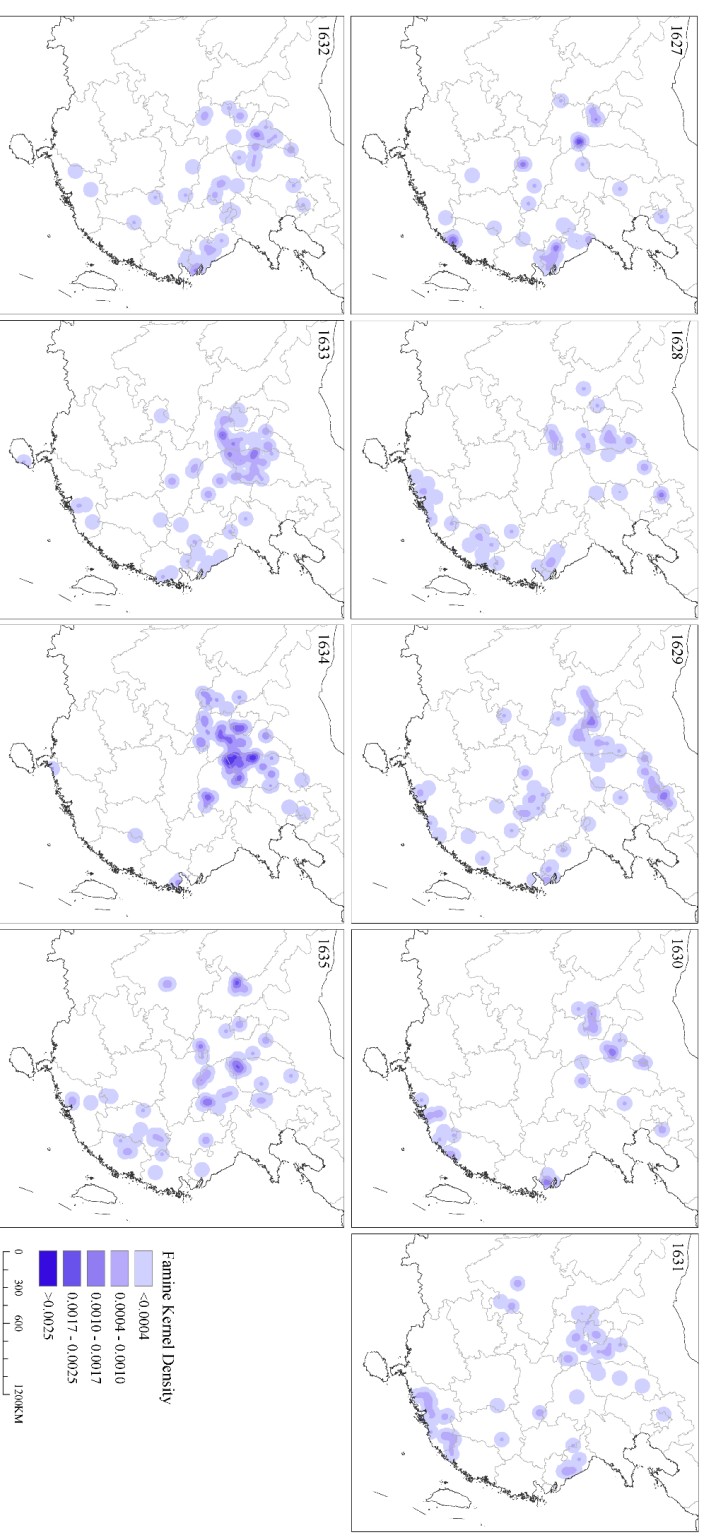

**Appendix D. Year-by-year spatial distribution of famine during 1627-1644**





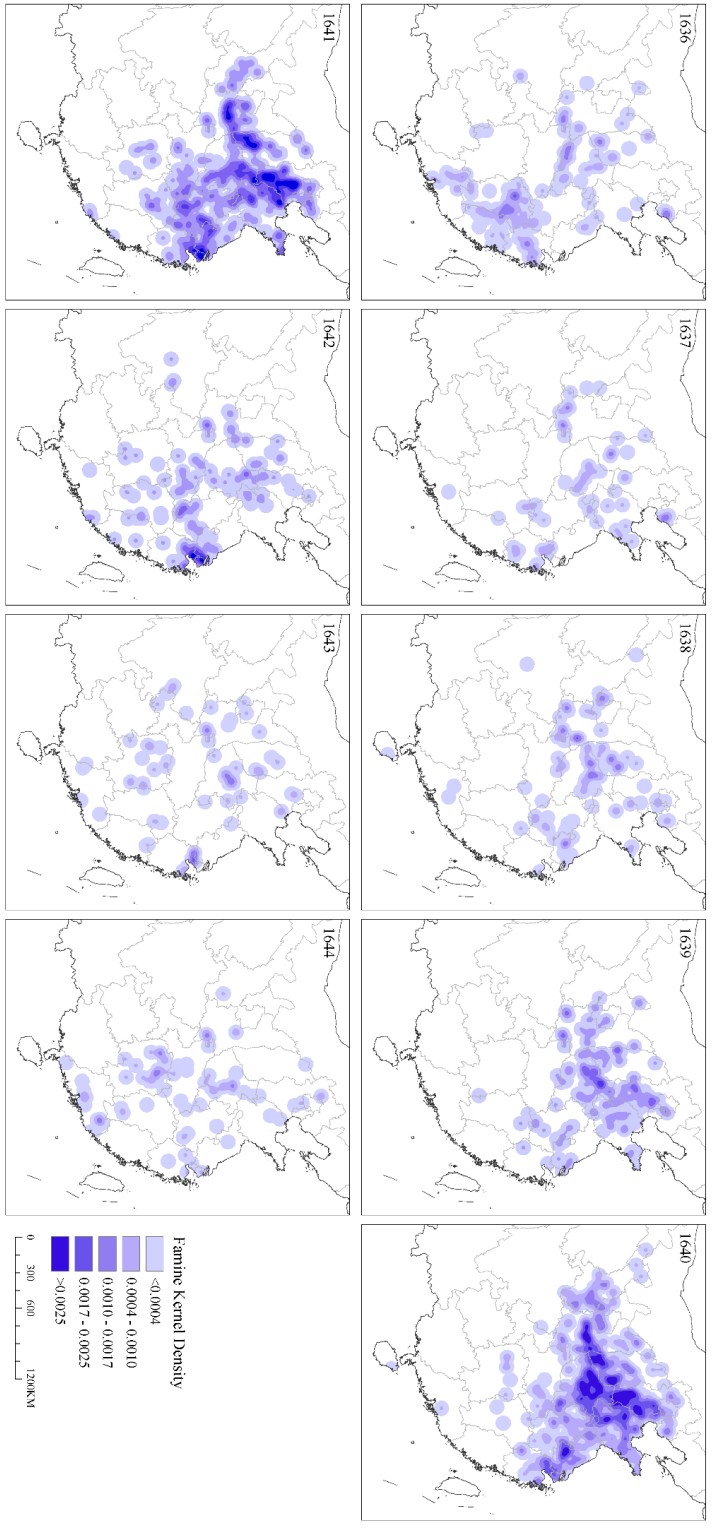