# Peer review of "The Spatial-Temporal Evolution of the Chongzhen Drought (1627-1644) in China and its Impact on Famine"

_Climate of the Past, 2024_

## Author Comment (AC2)

Thank you for dedicating your time and expertise to review our paper. Your comments and feedback are invaluable to enhancing the quality and clarity of the work. We are truly grateful for your careful review and comprehensive suggestions. Please see the responses below. Referee's comments (RC) are marked in bold font, authors' responses (AC) are marked in normal font. All line numbers mentioned correspond to the preprint version.

**Specific comments:**

**Line 39-49: I understand that the authors intend to begin with the topic of famine and subsequently introduce drought as one of its primary causes. However, in my view, the second paragraph of the introduction would benefit from a more detailed discussion of drought, especially since the first paragraph focuses solely on climate change. A logical progression from climate change to drought and then to famine would seem to flow more naturally and align more coherently with the subsequent narratives.**

Response: We agree with the logical progression suggested by the reviewer, which is indeed more conducive to paragraph articulation. Thus, we will reorganize this paragraph. Text:

*Drought, characterized as an extreme climatic event, may intensify the conflicts between humans and the environment at different time scales, influencing the trajectory of civilizational development. Prolonged droughts contributed to the collapse of the Classic Maya civilization (Medina-Elizalde and Rohling, 2012; Douglas et al., 2015), the migration of the Anasazi population (Benson et al., 2007), and the demise of Angkor, the capital of the Khmer Empire (Buckley et al., 2014). In China, drought is the most frequent natural disaster, with 1,074 years of recorded droughts from 1766 BCE to 1937 CE (Li et al., 2003; Deng, 2012). In historical times when agricultural harvests depended heavily on climatic conditions, long-lasting and widespread drought events declined food production and thus were likely to trigger famine (Teng et al., 2014). Defined as a state of extensive hunger resulting from a lack of food, famine denotes a crisis in food security. Famines may further lead to consequences like displacement, plague outbreaks, and social unrest. Historically, the large-scale peasantry uprisings in China in the late Eastern Han (180s), late Sui (610s), late Tang (860-880s), late Ming (1620-1640s), and late Qing dynasties (1840-1860s, 1890-1900s) all erupted in the context of extreme drought and famines (Ge, 2011; Fang et al., 2019), most of which played a critical role in dynasties changing. In sum, famine is a significant manifestation of the adverse effects of climate change reaching the human system. It also serves as a vital link in the chain of transmission of these effects to the economic, political, and military domains, which is particularly evident in agrarian societies.*

References:

Benson, L., Petersen, K., Stein, J.: Anasazi (pre-Columbian native-American) migrations during the middle-12th and late-13th centuries -were they drought induced?, Clim. Change, 83(1), 187-213, https://doi.org/10.1007/s10584-006-9065-y, 2007.

Buckley, B.M., Fletcher, R., Wang, S.Y.S, Zottoli, B., and Pottier, C.: Monsoon extremes and society over the past millennium on mainland Southeast Asia, Quaternary Science Reviews., 95(7), 1-19, https://doi.org/10.1016/j.quascirev.2014.04.022, 2014.

Deng, Y., The History of famine relief in China (in Chinese), Wuhan University Press, Wuhan, China, 2012.

Douglas, P.M., Pagani, M., Canuto, M.A., Brenner, M., Hodell, D.A., Eglinton, T.I., and Curtis, J.H.: Drought, agricultural adaptation, and sociopolitical collapse in the Maya Lowlands. Proc. Natl. Acad. Sci. U.S.A., 112(18), 5607-5612. https://doi.org/10.1073/pnas.1419133112, 2015.

Li, M., Li, S., Li, Y.: Studies on drought in the past 50 years in China (in Chinese with English abstract), Chinese Journal of Agrometeorology, 24(1), 7-10, 2003.

Medina-Elizalde, M., and Rohling, E.J.: Collapse of Classic Maya civilization related to modest reduction in precipitation, Science, 335(6071), 956-959, https://doi.org/10.1126/science.1216629, 2012.

Teng, J., Su, Y., and Fang, X.: The reconstruction and analysis of famine sequence from the Western Han to the Qing Dynasty (206BC~1911AD) (in Chinese with English abstract), Journal of Chinese Historical Geography, 29(4), 26-32, 2014.

Ge, Q.: Climate change in China's past dynasties (in Chinese), Science Press, Beijing, China, 2011.

Fang, X., Su, Y., Zheng, J., Xiao, L., Wei, Z., and Yin, J.: The Impact of Historical Climate Change on China's Social Economy (in Chinese), Science Press, Beijing, China, 2019.

**Line 46: It would be beneficial to clearly mark the corresponding periods of the late Eastern Han, Western Jin, late Sui, late Tang, late Ming, and late Qing dynasties in order to assist readers unfamiliar with Chinese history in understanding the context. The same applies to line 51.**

Response: We will add the corresponding periods of the peasantry uprisings mentioned in line 46, as well as the duration of the Ming dynasty in line 51. Text:

*Historically, the large-scale peasantry uprisings in China in the late Eastern Han (180s), late Sui (610s), late Tang (860-880s), late Ming (1620-1640s), and late Qing dynasties (1840-1860s, 1890-1900s) all erupted in the context of extreme drought and famines (Ge, 2011; Fang et al., 2019), most of which played a critical role in dynasties changing.*

*1627-1644 CE saw an extraordinary and extreme drought in China, known as the "Chongzhen Megadrought" because it coincided with the last period of the Ming Dynasty (1368-1644 CE), the Chongzhen Emperor's reign.*

References:

Ge, Q.: Climate change in China's past dynasties (in Chinese), Science Press, Beijing, China, 2011.

Fang, X., Su, Y., Zheng, J., Xiao, L., Wei, Z., and Yin, J.: The Impact of Historical Climate Change on China's Social Economy (in Chinese), Science Press, Beijing, China, 2019.

**Figure 1: The figure contains a mix of historical and modern geographical concepts, potentially leading to confusion. For instance, under contemporary geographical categorizations, Shanxi Province is not typically considered part of the northwest region, nor is Henan part of the North China region, and Hunan is not included in the Southeast region, among others. It might be clarifying to substitute "Northwest China" with "Northwest region" and "North China" with "Northern region." The former terms are commonly employed in the delineation of modern China's geographical divisions: East China, South China, North China, Central China, Southwest China, Northwest China, and Northeast China.**

Response: In this manuscript, the division of the study area does not strictly adhere to modern China's geographical categorizations.  Instead, we consider two aspects when dividing: firstly, the geographical attributes encompassing topography and climate, as well as the socio-economic distinctions; and secondly, the progression of droughts from 1627 to 1644.  Considering the spatial concurrence and temporal synchronicity of droughts, the division aims to minimize intra-regional differences while maximizing inter-regional differences.  Nonetheless, our naming of regions might lead to potential confusion with the modern geographical divisions of China, as the reviewer put forward.  Therefore, we accept the suggestion to substitute "Northwest China" with "Northwest region" and "North China" with "Northern region", to mitigate ambiguity and ensure a greater degree of nomenclatural consistency across the five regions.  Corresponding adjustments will be made in Figure 1, as well as in the text, tables, and other figures.

**Table 1: I have the following three types of questions:**

**(1) The translation of Chinese expression "崇祯十年旱" should be "Drought in the tenth year of Chongzhen period" which is analogous to the expression "Drought in the third year of Chongzhen period (崇祯三年旱)". Given this similarity, both records should be classified at the same level. However, in Table 1, the former is classified as level 3, while the latter is classified as level 2.**

**(2) The expression of "Drought in the third year of Chongzhen period (崇祯三年旱)" means "There is a drought in a specific year". "Winter drought (冬旱)" is also a situation in which drought occurs in a given year. From this perspective, the two records should be classified at the same level. However, in Table 1, the former is classified as level 2 and the latter is classified as level 1.**

**(3) How should readers interpret and compare different types of drought expressions? For instance, why is the drought level of 'Drying up of wells' classified as Level 3 in Table 1, which is higher than that of 'The grass and trees are withered and scorched', classified as Level 2?**
**It is recommended to provide more detailed descriptions on how to classify the original records using the semantic difference method. This would assist readers in achieving similar classification results with the same data. The authors could consider categorizing all descriptions into groups such as drought descriptions, plant descriptions, and descriptions of rivers, wells, and groundwater. Subsequently, they could summarize the text descriptions within each category and classify them into levels 1-4. The same approach could be applied to Table 2.**

Response:

(1) There is a clerical mistake in cell Level-3-2) in this table. The original Chinese text for "Great drought in the tenth year of Chongzhen period" should be "崇祯十年大旱". It differs from the record "Drought in the third year of Chongzhen period (崇祯三年旱)" in the intensity of drought. That's why they are classified at different levels. We will correct the mistake.

(2) Climate disaster records in Chinese historical documents exhibit two characteristics: focusing on anomalous rather than normal phenomena and emphasizing events that significantly affect agricultural production and the human-living. Thus, more attention is given to summer and autumn when recording drought. Compared to winter and spring, droughts in these seasons represent greater precipitation anomalies and are more likely to threaten agricultural production. Generally, if a drought occurs only in the winter, it would not be recorded as "drought in year XX."

In our database, some years and locations have records from multiple sources that can serve as evidence. For example, for the year 1639 in Lianshui County, Jiangsu Province, *A compendium of Chinese meteorological records of the last 3000 years* notes the event as "drought," while *Climatic historical materials for the last 500 years in East China* records it as "summer drought," among many similar instances.

Thus, we consider "Drought in a certain year" to represent droughts occurring in the summer or autumn and should be divided as level 2. In contrast, "winter drought" represents less precipitation anomalies and is categorized as level 1.

(3) Drought initially originates in the atmosphere, characterized by prolonged periods without precipitation or reduced rainfall, i.e., meteorological drought; then it begins to impact other Earth system spheres. In formulating the criteria for drought classification in this study, we gave priority to those records directly related to the season of occurrence, duration, and intensity of meteorological drought. Additionally, other relevant records are also considered, such as hydrological conditions including declines in river levels, drying of surface water bodies, and lowering of groundwater levels. Similarly, vegetation conditions such as plant wilting or even widespread plant mortality are referenced. Based on these phenomena, the drought level is determined, as illustrated in the table below.

|  | Seasons, duration, and intensity of drought | Hydrological conditions | Vegetation Conditions | Comprehensive descriptions of drought conditions |
|---|---|---|---|---|
| Level-1 | Drought in the dry season and no drought in the wet season, or one-month drought. | - | - | - |
| Level-2 | Drought in one wet season. | The river and lake levels have dropped significantly, but have not dried up completely. | Plant wilting | "Drought" occurred in a certain year is recorded |
| Level-3 | Drought in two wet seasons, or severe drought in one wet season. | Complete drying up or ceasing to flow of rivers and lakes, or lowering of groundwater levels. | Extensive plant mortality | "Great drought" occurred in a certain year is recorded |
| Level-4 | Drought throughout the year, or severe drought in two wet seasons. | - | - | - |

As for the example the reviewer mentioned, well water originates from underground aquifers, while plants primarily absorb capillary water in the soil through their roots. The burial depth of the former exceeds that of the latter. Moreover, the phenomenon of "drying up of wells" indicates a significant decline in the groundwater level, whereas the withering of plants (not as severe as the widespread death of plants) suggests a reduction in soil moisture. Consequently, the degree of drought represented by the "drying up of wells" should be considered greater than that indicated by "the grass and trees are withered and scorched."

**Line 149: Readers unfamiliar with China may not know the locations of the Wei River, Fen River, and Guanzhong Plain. The same applies to Line 189, where the locations of Changsha, Hexi Corridor, Fen and Zhang Rivers in Shanxi, Wen and Si Rivers in Shandong, Baiyangdian, Suzhou Creek, etc., have not been introduced in the article.**

Response: Thanks for the suggestion. We decide to add another map to illustrate the geographical overview of the study area as Figure 1(b), shown below. This map will label the rivers and lakes mentioned in the text, such as the Wei River, Fen River, Zhang River, Dawen River, Si River, Suzhou Creek, and Baiyangdian. Additionally, we will remove or replace those terms in the manuscript that may be less understandable to readers, such as "Guanzhong Plain," "Changsha," and "Hexi Corridor." For instance, the phrase in line 149, "the area along the Yellow, Wei, and Fen Rivers in Guanzhong Plain and southwestern Shanxi," will be revised to "the area along the Yellow, Wei, and Fen Rivers in central Shaanxi and southwestern Shanxi."

[Figure]

Figure 1: Map of the study area
(a) The location of the study area and subregions; (b)DEM of the study area with main rivers and lakes

**Figure 2a: What distinguishes a Drought Zone from a Severe Drought Zone? What criteria are used to classify areas into Drought Zones and Severe Drought Zones? Is it the case that areas experiencing level 1-2 drought events are classified as Drought Zones, while areas experiencing level 3-4 drought events are classified as Severe Drought Zones? The same question applies to the classifications in Figure 4a.**

Response: The classification of Drought Zone and Severe Drought Zone was based on the drought kernel density distribution map for the entire study period (1627-1644). We used the natural breaks method to classify kernel densities, with densities from 0.00008 to 0.00392 representing Drought Zone, and densities from 0.00392 to 0.02084 representing Severe Drought Zone. Thus, there is not a direct correspondence between this classification and the levels of drought events. However, the kernel density of a given raster is determined by the number and levels of drought events surrounding it.

To avoid ambiguity, we will mark the ranges of drought kernel densities corresponding to the Drought Zone and Severe Drought Zone in the legend of Figure 2a, shown below. Similar modifications will also be made to Figure 4a.

[Figure]

Figure 2: Spatial and Temporal Patterns of Drought, 1627-1644  (a) Overall spatial distribution of drought, 1627-1644. (b) Drought Kernel Density Index (DKDI) series for five regions

**Figure 2b: I have the following two suggestions.**

**(1) By converting the ordinates to the same scale, the changes and differences in the DKDI index across each region can be displayed more intuitively. For example, it can be unified into a range from 0 to 1 with an interval of 0.1.**

**(2) The expression of the start and end time of each stage in the figure is different from the text content. Taking the "Starting Phase" as an example, the start time seems to be someday before 1627 and the end time seems to be someday after 1633 from the figure.**

Response: We accept the suggestion to standardize the vertical axis of Figure 2b to the same scale, ranging from 0 to 1 with an interval of 0.2. We will also adjust the plotting expression for phases to avoid ambiguity, as shown in Figure 2b above.

**Line 160-170: The changes of DKDI in the southeast and southwest regions are not mentioned in this paragraph.**

Response: Regarding the changes of DKDI in the Southeast and Southwest regions, we have discussed this in the previous paragraph (see lines 157-159 in the preprint). Compared to the other regions, these

two regions have overall low DKDI values, with peaks not exceeding 0.2, indicating that they were not the main drought-affected regions. Therefore, a detailed analysis is not conducted.

**Line 171: It is recommended to provide more details on how the four phases of the drought were determined by analyzing the temporal and spatial variations of the drought.**

Response: The four phases were determined based on the temporal changes in DKDI and the comparison of DKDI across the three main regions (Northwest, Northern, and Jianghuai regions). We will add a sentence in line 179 to clarify the key events that led to the division into four phases. Text:

*The delineation of the four phases is informed by key events: declining DKDI in the Northwest Region while increasing DKDI in the Northern Region in 1634; the beginning of a significant increase in DKDI in all three regions mainly affected in 1638; and a remarkable decline and following stabilization in DKDI in those three regions in 1642.*

**Line 190: This line implies that the Yellow River once flowed through Jiangsu Province. But as we can see from Figure 1, the Yellow River does not flow through Jiangsu Province. This discrepancy is due to Figure 1 representing the modern course of the Yellow River, which has historically changed its course. This statement could potentially cause confusion among readers and therefore necessitates additional clarification.**

Response: We will add a footnote at line 190 to explain. Text:

*In the 17th century, the Yellow River flowed through Jiangsu Province into the sea. However, after a breach at Tongwagang, Henan Province, in 1855, its course changed. Currently, it does not flow through Jiangsu Province, as illustrated in Figure 1.*

**Figure 4b: Can it be divided into distinct phases like drought? If not, the phases of drought are suggested to be re-marked on the chart for easier comparison.**

Response: To avoid confusion with the drought phases, we have not segmented the famine evolution into phases. However, we will mark the drought stages above the FKDI series graph in Figure 4b, as shown below, to facilitate the analysis of famine evolution during different phases.

[Figure]

Figure 3: Spatial and Temporal Patterns of Famine, 1627-1644   (a) Overall spatial distribution of famine, 1627-1644. (b) Famine Kernel Density Index (FKDI) series for five regions

**Table 4: What is the correlation coefficient in the fourth year? If we want to draw conclusions that affect three years, shouldn't we at least list the correlation coefficient for the fourth year?**

Response: Thanks for the referee's suggestion, which prompted us to revisit the correlation analysis in Table 4. We found that these results do not significantly contribute to the paper and do not provide conclusions different from those in Table 3. Therefore, we will remove this table and its related content. Thus, Section 5 will now include: (1) correlation analysis at the regional scale to demonstrate the correlation between DKDI and FKDI, and the continuity of drought's impact on famine; (2) regression analysis at the sub-provincial scale to explore the varying contributions of drought to famine across different regions.

**Line 302: Do these 15 response types occur exclusively in the Yangtze-Huai Region, or are similar patterns observed in other regions as well?**

Response: Measures in Table 5 were recorded across various regions, but the Jianghuai Region exhibited the greatest frequency of those measures, particularly at the local government level. In the Northern and Northwest Regions, the recorded response measures predominantly involve financial or food relief, also with some instances of giving porridge and selling grain at low prices. However, measures like treatment, corpse management, hosting of orphans, and praying for rain are rare. There are two possible reasons for this: compared to other regions, the Jianghuai Region owned a more

developed economy and more abundant grain storage, providing a solid material basis for disaster response; additionally, the local officials in the Jianghuai Region showed stronger governance capabilities. Many of the measures, such as treatment and corpse management, were initiated by those officials, and the records documenting their names. It illustrates that human response measures can moderate the transmission process of drought impacts and reduce social damage. Due to space limitations, we choose the most typical region, Jianghuai Region, as an example for illustration.

**Technical corrections:**

**Line 139: Formulas need to be numbered.**

Response: We will number the formulas in the manuscript.

**Figure 3: The graticules appear to be missing from the figure. The same issue is observed in Figure 6.**

Response: We will add the graticules in Figure 3 and Figure 6, as shown below.

[Figure]

Figure 4: Spatial distribution of drought during representative years of each phase

[Figure]

Figure 5: Areas affected by the locust infestation and plague

(a) Locust infestation (b)Plague

**Figure 4: Typically, figures are not placed directly beneath the title.**

Response: We will move Figure 4 below the first two paragraphs of Sect 4.

**Table 6: This table should be reformatted into a three-line table.**

Response: We will reformat Table 6 into a three-line table, as shown below.

**Table 6: Response measures in the Yangtze-Huai Region from 1627 to 1644**

| | Response Measure | Main Actor | Meaning |
|---|---|---|---|
| (1) | Reseeding | Farmer | After the crop dies due to drought, re-plant some if rainfall occurs. |
| (2) | Locust catching | Farmer | Catch locusts to prevent them from destroying crops. Sometimes local government also encouraged people to do so and gave them some grain or money as an award. |
| (3) | Food substitution | Famine victim | Eat wild herbs, chaff, grass roots, tree bark, soil, and so on to satisfy their hunger. The most extreme case is cannibalism (i.e. some people kill others and eat corpses). |
| (4) | Selling property | Famine victim | Selling property such as houses, land, cattle, agricultural tools, etc., in exchange for money to buy food. The most extreme case is trafficking women and children. |
| (5) | Displacement | Famine victim | Leave their hometown and flee to surrounding areas in search of food |
| (6) | Robbery | Famine victim | Rob on the road, or rob the homes of the wealthy, landlords, and gentries to obtain food or money. |
| (7) | Giving porridge | Local government, officials, gentries | Open porridge factories to feed famine victims |

| (8) | Selling grain at low price | Local government | Sell stored grain at low prices to prevent excessive increases in grain prices on the market due to shortages. |
|---|---|---|---|
| (9) | Financial or food relief | Local government, central government | Distribute food or money to the victims directly. |
| (10) | Tax exemption | Central government | A discretionary exemption from taxes or corvees. |
| (11) | Tax substitution | Central government | Allow disaster areas to convert the tax grain into money or other items to hand in. |
| (12) | Treatment | Doctors, local government | Distribute medicine to the population if a plague occurred, or hire doctors to treat the patients. |
| (13) | Corpse management | Local government, officials, gentries | Distribute coffins to the families of the deceased, or bury the corpses together. |
| (14) | Hosting of orphans | Local government | Set up shelters to take in abandoned kids. |
| (15) | Praying for rain | Local officials | Offer sacrifices to gods and pray for rain. |

**Figure 7: The figure is not cited in the text.**

Response: We will add the citation of Figure 7 in line 347.

---

## Author Comment (AC3)

We sincerely appreciate your time and expertise in reviewing our paper. Your comments are helpful, facilitating us to improve our work. We are also grateful for your insightful suggestions. Please find our responses below. Referee's comments (RC) are marked in bold font, while authors' replies (AC) are in regular font. All line numbers mentioned correspond to the preprint version. Thanks again!

1. **In the discussion section, the analysis of the impact of other factors on the famine appears somewhat superficial. It could be augmented by considering the "Three Extra Levies" (special taxes imposed for military expenses – the Liao levy: imposed in response to the war with Manchuria——later the Qing dynasty, Suppression levy: imposed in response to large-scale peasant uprisings, and Training levy: imposed for the training of new armies), which intensified the burden on peasants and influenced the famine.**

Reply: Thanks for your suggestion. We will add the "Three Extra Levies" as the fifth factor at line 340. Text:

*(5) The increasing taxes. Due to the Manchu invasion and peasant uprisings, the Ming government imposed additional taxes to meet the escalating military expenditures. The most notable examples were the "Three Extra Levies": the Liao levy (initiated in 1618 for the war in the northeast), the Suppression levy (initiated in 1637 to quell peasantry uprisings), and the Training levy (initiated in 1639 for training the new army). By 1639, the total amount of these additional levies even exceeded the regular tax revenue of normal years. These increasing taxes further lowered the living standards of the populace and exacerbated social conflicts.*

**2. In Figure 1, important locations mentioned in the text should be marked on the map to facilitate reader comprehension. Examples include Guanzhong and the Weihe River.**

Reply: We will add another map to illustrate the geographical overview of the study area as Figure 1(b), shown below. And those important locations, especially rivers and lakes, will be marked here. At the same time, we will remove or replace those terms in the manuscript that may still be confusing. For example, we use "central Shaanxi" to replace "Guanzhong Plain" at line 149, so that readers can refer to Figure 1(a) for the location.

[Figure]

**Figure 1: Map of the study area**
**(a) The location of the study area and subregions; (b)DEM of the study area with main rivers and lakes**

3. **In Figure 3, the first two small graphs require certain modifications. The maps for 1633 and 1636 should illustrate the average drought index during the initial phase (1627-1633) and the developmental phase (1634-1637) mentioned in the text, for clearer presentation.**

Reply: Thanks for the comment, making us think about this issue carefully. We attempted to create a map illustrating the average drought kernel density index for each phase. However, we found it to be of little help in displaying spatial variability in the drought zone. The primary reason is the differences in the spatial extent of drought from year to year, even within the same phase. Such variability would be diluted in an average kernel density map. For instance, the map below shows the average drought kernel density during the Starting Phase. It looks as if most of the study area experienced mild droughts. In reality, specific areas suffered severe droughts, such as Hebei Province in 1628 and Shaanxi and Shanxi Provinces in 1633, while most areas experienced drought only in a single year.

[Figure]

**Figure: The average drought kernel density in the starting phase**

Therefore, we opted to maintain the original mapping approach, i.e., to present one representative year from each of the Starting, Transition, and Receding Phases, and to display the distribution of droughts for each year during the Peak Phase. Readers who wish to get a more complete picture of the spatial variability of drought are referred to the year-by-year spatial distribution maps provided in Appendix B.

---

## Author Comment (AC4)

Thank you very much for your insightful comments. We have carefully considered your feedback, which helps us a lot to improve the work. We hope that our replies and additional information will address your concerns and contribute to a more comprehensive understanding of the issues discussed in our study. Please find the detailed response below. Referee's comments (RC) are marked in bold font, while authors' replies (AC) are in regular font. All line numbers mentioned correspond to the preprint version. Thanks again!

**1/ General questions:**

**Insofar as famine does not affect all regions at the same time, aren't subsistence crises partly "exchange-soluble"? In other words, are there attempts (by local or central government, farmers...) to alleviate famines - on a large or small scale - through massive imports from other regions, through trade between upstream and downstream regions, or through smuggling? Similarly, is there any legislation against trade in times of crisis?**

Reply: Regional interaction plays a crucial role in addressing famine. However, during 1627-1644, regional interaction mainly manifested through the form of human migration, wherein victims fled their hometowns and migrated to surrounding areas in search of food (measure (5) in Table 6).

As for the transfer of grain, there are a few documented instances, such as in Ankang County, Shaanxi Province, where "Liu Yingke returned with rice from Hubei Province by boat and dispersed it（刘盈科自楚贩米归，倾舟散之）". However, such records are rare. Conversely, numerous records said 'no place to buy grain (无籴处)', suggesting that even those with financial means could not procure grain from other areas.

Several factors contributed to this situation: (1) The study period coincided with the end of the Ming Dynasty, characterized by a widespread shortage of grain storage across the country. Additionally, the central government's diminished governance capacity hindered long-distance food transportation on a national scale. (2) Adjacent regions often suffer from droughts and famines at the same time, especially in the peak phase, making it challenging for them to achieve self-sufficiency, let alone assist neighboring regions. (3) The Chongzhen period was marked by social unrest, with numerous peasant uprisings and famine victims turning to banditry. These groups blocked roads and looted food, causing great obstacles to the transport of grain.

Given this situation, the government did not prohibit trade but rather encouraged it. For example, the governor of Shaanxi, Sun Chuanting, ordered soldiers to clear the roads to facilitate rice purchasing and encouraged people to traffic grain to Hanzhong on their own. However, the scale and effect of such measures were limited, and overall, they contributed little to addressing the famine.

References:

Zhang, D.: A compendium of Chinese meteorological records of the last 3000 years (in Chinese), Phoenix Publishing House, Nanjing, China, 2004.

Sun, C.: Sun Chuanting's documents: Memorial to the throne on the money and grain of Hanzhong (in Chinese), Zhejiang People's Publishing House, Hangzhou, China, 1983.

**Table 6 (line 8) shows food distributions (also mentioned on line 306): do local governments own granaries to prevent food crises (which presupposes purchases and infrastructure) or do they legally take control of the trade and grain stored by rural communities?**

Reply: Measures (7) to (9) in Table 6 were essentially to distribute grain to famine victims. These grains were predominantly sourced from warehousing storage, which in the Ming Dynasty included various types such as disaster-preparing granaries (预备仓), price-stabilizing granaries (常平仓), community granaries (社仓), and charity granaries (义仓). The first two types were constructed and managed by local governments, while the latter two were established and operated by civilian groups, with the government playing a supervisory role. The grain stored in these facilities was sourced from agricultural taxes, grain purchased in bountiful years, and criminal fines.

In addition to warehousing, donations played a significant role, as officials and wealthy gentry contributed money or grain during disaster years, which was then used for distributing porridge, stabilizing grain prices, or providing direct relief. Given the crucial role of donations during Chongzhen Drought, we will include it in Table 6 as measure (10), as follows:

**Table 1: Response measures in the Yangtze-Huai Region from 1627 to 1644**

|  | Response Measure | Main Actor | Meaning |
|---|---|---|---|
| (7) | Giving porridge | Local government, officials, gentries | Open porridge factories to feed famine victims |
| (8) | Selling grain at low price | Local government, gentries | Sell stored grain at low prices to prevent excessive increases in grain prices on the market due to shortages. |
| (9) | Financial or food relief | Local government, central government | Distribute food or money to the victims directly. |
| (10) | Donation | Officials, gentries | Donate money or grain to support measures (7)~(9). |

References:

Chen, G.: Research on the warehouse system and famine preparation in the Ming Dynasty (in Chinese), Seeker, (5), 1991.

Ju, M., and Yi, L.: The change and new understanding of ever-normal granary in the Ming Dynasty (in Chinese with English abstract). Journal of Henan Normal University (Philosophy & Social Sciences), 49(5), 129-135, 2022.

**2/ Specific questions:**
**In table 6, line 15, "Praying for rain": is this a political and religious initiative, or just an individual and local one? Are there any general incentives that might underline the level of social stress?**

Reply: The records of praying for rain identify three types of actors: (1) the Emperor, for example, "In the summer (of 1633), there was a severe drought. The Emperor pardoned prisoners and prayed in the southern suburbs, after which heavy rain fell (夏大旱, 清理狱囚, 上步祷南郊, 回銮, 大雨)"; (2) central officials, for example, "On the day of Gui Si in the fourth month (of 1628), the Emperor ordered the Ministry of Rites to pray for rain (四月癸巳，谕礼部祷雨)"; (3) local officials, for example, in Jiangyin County, Jiangsu Province, "There was no rain from the 5th to the 7th months (of 1640). Zhang Sijia, in charge of coastal defense, initiated prayer with his subordinates. Zhang Fenghe, the local education commissioner, also participated in the prayer (夏五月不雨至七月，海防张嗣嘉率属步祷，学台张凤翙亦出祷)."

These examples demonstrate that praying for rain was a political act. Influenced by the Confucian theory of "interactions between heaven and mankind", the government intended to show concern for droughts and self-reflection by praying. This practice helped to appease the populace and mitigate tensions to some extent. However, in the case of Chongzhen Drought, the praying records are sparse and lack detailed descriptions of cause and effect, making further analysis difficult.

**The authors mentioned "cannibalism" in Table 2 and Table 6 and state at line 317 that "Severe famine swept through the region, and there were even instances of cannibalism." Are these extreme cases sufficiently well-documented and numerous to be used in the classification scale of famine levels and response measures, or are they local epiphenomena?**

Reply: From 1627 to 1644, there were 557 records of "cannibalism", which is a large number. Over 80% of these events took place during the peak phase of drought (1638-1641). Spatially, such events happened over a wide range, but 88% concentrated in the Northwest and Northern Regions.

Cannibalism signifies that the famine has reached an extreme level of severity, reflecting both the extreme scarcity of food and the collapse of ethics and social order. Based on the records, it is evident that cannibalism, as the most extreme form of food substitution, occurred during the study period. The quantity and content of these records support their use as one criterion for classifying famine levels. However, we also believe that such extreme events should be treated with great caution. We only took them as evidence of famine without conducting further specific analyses of themselves.

**As RC2 pointed out, important locations mentioned in the text should be marked on the map (Figure 1) to facilitate reader comprehension.**

Reply: Thanks a lot for your comment, this is something we overlooked while writing the paper. We will redesign Figure 1 to better present the geographical overview of the study area, as shown below. And those important locations will be marked there. At the same time, we will remove or replace those terms in the manuscript that may still be confusing. For example, we use "central Shaanxi" to replace "Guanzhong" at line 149, so that readers can refer to Figure 1(a) for the location.

[Figure]

**Figure 1: Map of the study area**
**(a) The location of the study area and subregions; (b) DEM of the study area with main rivers and lakes**

---

## Author Response (AR1)

Dear editors and reviewers,

Thank you for dedicating your time to reviewing our manuscript. Your comments and feedback have been invaluable in improving the quality of our paper. Based on the comments, we have made significant modifications and updated the manuscript. Please find the itemized replies below. The referees' comments (RC) are marked in bold font, while the authors' replies (AC) are in regular font. In the document named "Modified manuscript", the revised sections are highlighted in red and correspond to the line numbers mentioned below. Thanks again!

**Community Comment (CC1)**

**Comment 1: Based on historical documents, there is already a widely used criterion for classifying drought-flood levels in the academic community (referred to *Yearly charts of dryness/wetness in China for the last 500-year period*). Given this, why did the authors choose to define a new grading criteria (Table 1) instead of employing the existing one?**

Reply: The criteria in *Yearly Charts of Dryness/Wetness in China for the Last 500-Year Period (中国近五百年旱涝分布图集)* (hereafter referred to as *"Yearly Charts"*) were defined to classify the level of drought and flood disasters. The criteria not only consider the hazard's duration and intensity but also the social consequences caused. Events such as famine, surges in grain prices, and population displacements are also considered as criteria for determining levels. However, one focus of this study is to investigate the interrelation between drought and famine, necessitating that data on droughts and famines be mutually independent. Therefore, the grading criteria in *"Yearly Charts"* are inapplicable for this study.

Employing a similar methodology, our proposed grading criteria were also based on semantic differences, maintaining an emphasis on the duration and intensity of drought events. However, a distinction lies in that our criteria specifically target drought manifestations within the natural environment, particularly meteorological and hydrological droughts, while intentionally excluding agricultural and socio-economic droughts that involve human intervention. Consequently, the spatial-temporal patterns of drought identified in this study reflect the anomaly of the natural system, making it possible for comparative analysis with the famine process, which represents the anomaly of the human system.

The criteria in "Yearly charts" are attached here for reference.

| | Criteria | Keyword examples |
|---|---|---|
| Level-1 Very wet | Long-lasting and intense precipitation, widespread flooding, catastrophic typhoon rains along the coast, etc. | Spring and summer rains; summer rain for more than a decade, the river overflowed; spring and summer floods, drowned people and animals without counting; summer and fall floods inundated seedlings; heavy rain for days, the land can be boat; several counties flood; hurricanes and heavy rains, drifted away the fields and houses. |
| Level-2 Wet | Sustained precipitation with little damage in spring and fall. Locally flood. Hurricanes and heavy rains with little damage. | Spring rains hurt the seedlings; fall rains hurt the crops; April floods, famine; August floods; a county's flash mountain flood, destroying the fields. |
| Level-3 Normal | The year was productive, great harvest, or no drought or flood can be recorded. | Great Harvest; fall Harvest; great year |

| | | |
|---|---|---|
| Level-4 Dry | Drought that causes little damage, lasting for a single season or month; localized drought. | Spring drought; autumn drought; drought; drought in a particular month; rains are scarce; drought and locust. |
| Level-5 Very dry | A drought that lasts for several months or spans seasons; a widespread severe drought. | Spring and summer drought, barren land, people eat grass roots and bark; summer and fall drought, the crops are withered; summer severe drought, famine; no rain from the 4th to 8th month, no harvest; rivers/ponds/wells dry up; great drought in Jiangnan region. |

**Comment 2: When classifying drought levels, how to deal with a drought event that spans years?**

Reply: The basic time unit for drought events and drought levels in this study is the year, while the basic spatial unit is the county. Drought events that span years are dissected into their seasonal components for detailed evaluation. For example, a "winter-spring drought" is considered to be a winter drought in the first year (level 1) and a spring drought in the second year (level 1).

In Sect.2.2 (lines 119-120), we added a sentence to clarify this. Text:

The basic time unit for drought events is the year, while the basic spatial unit is the county.

**Comment 3: The maps and names of the provinces used in the manuscript are modern, but historical place names and provincial divisions may have differed. I suggest explaining how to convert location information with examples.**

Reply: As mentioned in Appendix A, the database for this study comprises 6,282 records, 5,006 of which come from "A compendium of Chinese meteorological records of the last 3000 years." This collection had already converted historical place names into corresponding modern ones during the compilation process. Thus, the majority of the records did not require any conversion of locations. However, for some records extracted from other collections that only provided historical place names, conversions to modern place names were carried out according to "The Historical Atlas of China (Volume 7)" (Tan, 1982). The belonging provinces were identified based on the modern administrative divisions of China, referencing the "Handbook of the People's Republic of China's Administrative Divisions" (Ministry of Civil Affairs of the People's Republic of China, 2020).

To clarify, we added an example in lines 98-101. Text:

For example, there is an item of record with the ancient place name Wuding, corresponding to modern Huimin County in Shandong Province. Hence, the "location" of this record is noted as Huimin County and the "province" as Shandong Province.

**Line 39-49: I understand that the authors intend to begin with the topic of famine and subsequently introduce drought as one of its primary causes. However, in my view, the second paragraph of the introduction would benefit from a more detailed discussion of drought, especially since the first paragraph focuses solely on climate change. A logical progression from climate change to drought and then to famine would seem to flow more naturally and align more coherently with the subsequent narratives.**

Reply: We agree with the logical progression suggested by the reviewer, which is indeed more conducive to paragraph articulation. Thus, we reorganized this paragraph (lines 40-53 in the manuscript). Text:

Drought, characterized as an extreme climatic event, may intensify the conflicts between humans and the environment at different time scales, influencing the trajectory of civilizational development. Prolonged droughts contributed to the collapse of the Classic Maya civilization (Medina-Elizalde and Rohling, 2012; Douglas et al., 2015), the migration of the Anasazi population (Benson et al., 2007), and the demise of Angkor, the capital of the Khmer Empire (Buckley et al., 2014). In China, drought is the most frequent natural disaster, with 1,074 years of recorded droughts from 1766 BCE to 1937 CE (Li et al., 2003; Deng, 2012). In historical times when agricultural harvests depended heavily on climatic conditions, long-lasting and widespread drought events declined food production and thus were likely to trigger famine (Teng et al., 2014). Defined as a state of extensive hunger resulting from a lack of food, famine denotes a crisis in food security. Famines may further lead to consequences like displacement, plague outbreaks, and social unrest. Historically, the large-scale peasantry uprisings in China in the late Eastern Han (180s), late Sui (610s), late Tang (860-880s), late Ming (1620-1640s), and late Qing dynasties (1840-1860s, 1890-1900s) all erupted in the context of extreme drought and famines (Ge, 2011; Fang et al., 2024), most of which played a critical role in dynasties changing. In sum, famine is a significant manifestation of the adverse effects of climate change reaching the human system. It also serves as a vital link in the chain of transmission of these effects to the economic, political, and military domains, which is particularly evident in agrarian societies.

**Line 46: It would be beneficial to clearly mark the corresponding periods of the late Eastern Han, Western Jin, late Sui, late Tang, late Ming, and late Qing dynasties in order to assist readers unfamiliar with Chinese history in understanding the context. The same applies to line 51.**

Reply: We added the corresponding periods of the peasantry uprisings mentioned in lines 48-50, as well as the duration of the Ming dynasty in line 55. Text:

Historically, the large-scale peasantry uprisings in China in the late Eastern Han (180s), late Sui (610s), late Tang (860-880s), late Ming (1620-1640s), and late Qing dynasties (1840-1860s, 1890-1900s) all erupted in the context of extreme drought and famines (Ge, 2011; Fang et al., 2024), most of which played a critical role in dynasties changing.

1627-1644 CE saw an extraordinary and extreme drought in China, known as the "Chongzhen Megadrought" because it coincided with the last period of the Ming Dynasty (1368-1644 CE), the Chongzhen Emperor's reign.

**Figure 1: The figure contains a mix of historical and modern geographical concepts, potentially leading to confusion. For instance, under contemporary geographical categorizations, Shanxi Province is not typically considered part of the northwest region, nor is Henan part of the North**

**China region, and Hunan is not included in the Southeast region, among others. It might be clarifying to substitute "Northwest China" with "Northwest region" and "North China" with "Northern region." The former terms are commonly employed in the delineation of modern China's geographical divisions: East China, South China, North China, Central China, Southwest China, Northwest China, and Northeast China.**

Reply: In this manuscript, the division of the study area does not strictly adhere to modern China's geographical categorizations. Instead, we consider two aspects when dividing: firstly, the geographical attributes encompassing topography and climate, as well as the socio-economic distinctions; and secondly, the progression of droughts from 1627 to 1644. Considering the spatial concurrence and temporal synchronicity of droughts, the division aims to minimize intra-regional differences while maximizing inter-regional differences.

Nonetheless, our naming of regions might lead to potential confusion with the modern geographical divisions of China, as the reviewer put forward. Therefore, we accept the suggestion to substitute "Northwest China" with "Northwest region" and "North China" with "Northern region", to mitigate ambiguity and ensure a greater degree of nomenclatural consistency across the five regions. Corresponding adjustments have been made in Figure 1, as well as in the text, tables, and other figures.

**Table 1: I have the following three types of questions:**

**(1) The translation of Chinese expression "崇祯十年旱" should be "Drought in the tenth year of Chongzhen period" which is analogous to the expression "Drought in the third year of Chongzhen period (崇祯三年旱)". Given this similarity, both records should be classified at the same level. However, in Table 1, the former is classified as level 3, while the latter is classified as level 2.**

**(2) The expression of "Drought in the third year of Chongzhen period (崇祯三年旱)" means "There is a drought in a specific year". "Winter drought (冬旱)" is also a situation in which drought occurs in a given year. From this perspective, the two records should be classified at the same level. However, in Table 1, the former is classified as level 2 and the latter is classified as level 1.**

**(3) How should readers interpret and compare different types of drought expressions? For instance, why is the drought level of 'Drying up of wells' classified as Level 3 in Table 1, which is higher than that of 'The grass and trees are withered and scorched', classified as Level 2?**
**It is recommended to provide more detailed descriptions on how to classify the original records using the semantic difference method. This would assist readers in achieving similar classification results with the same data. The authors could consider categorizing all descriptions into groups such as drought descriptions, plant descriptions, and descriptions of rivers, wells, and groundwater. Subsequently, they could summarize the text descriptions within each category and classify them into levels 1-4. The same approach could be applied to Table 2.**

Reply:

(1) There is a clerical mistake in cell Level-3-2) in this table. The original Chinese text for "Great drought in the tenth year of Chongzhen period" should be "崇祯十年大旱". It differs from the record "Drought in the third year of Chongzhen period (崇祯三年旱)" in the intensity of drought. That's why they are classified at different levels. We have corrected the mistake.

(2) Climate disaster records in Chinese historical documents exhibit two characteristics: focusing on anomalous rather than normal phenomena and emphasizing events that significantly affect agricultural production and the human-living. Thus, more attention is given to summer and autumn when recording drought. Compared to winter and spring, droughts in these seasons represent greater precipitation anomalies and are more likely to threaten agricultural production. Generally, if a drought occurs only in the winter, it would not be recorded as "drought in XX year."

In our database, some years and locations have records from multiple sources that can serve as evidence. For example, for the year 1639 in Lianshui County, Jiangsu Province, A compendium of Chinese meteorological records of the last 3000 years notes the event as "drought," while Climatic historical materials for the last 500 years in East China records it as "summer drought," among many similar instances.

Thus, we consider "Drought in a certain year" to represent droughts occurring in the summer or autumn and should be divided as level 2. In contrast, "winter drought" represents less precipitation anomalies and is categorized as level 1.

(3) Drought initially originates in the atmosphere, characterized by prolonged periods without precipitation or reduced rainfall, i.e., meteorological drought; then it begins to impact other Earth system spheres. In formulating the criteria for drought classification in this study, we gave priority to those records directly related to the season of occurrence, duration, and intensity of meteorological drought. Additionally, other relevant records are also considered, such as hydrological conditions including declines in river levels, drying of surface water bodies, and lowering of groundwater levels. Similarly, vegetation conditions such as plant wilting or even widespread plant mortality are referenced. Based on these phenomena, the drought level is determined, as illustrated in the table below.

| | Seasons, duration, and intensity of drought | Hydrological conditions | Vegetation Conditions | Comprehensive descriptions of drought conditions |
|---|---|---|---|---|
| **Level-1** | Drought in the dry season and no drought in the wet season, or one-month drought. | - | - | - |
| **Level-2** | Drought in one wet season. | The river and lake levels have dropped significantly, but have not dried up completely. | Plant wilting | "Drought" occurred in a certain year is recorded |
| **Level-3** | Drought in two wet seasons, or severe drought in one wet season. | Complete drying up or ceasing to flow of rivers and lakes, or lowering of groundwater levels. | Extensive plant mortality | "Great drought" occurred in a certain year is recorded |
| **Level-4** | Drought throughout the year, or severe drought in two wet seasons. | - | - | - |

As for the example the reviewer mentioned, well water originates from underground aquifers, while plants primarily absorb capillary water in the soil through their roots. The burial depth of the former exceeds that of the latter. Moreover, the phenomenon of "drying up of wells" indicates a significant decline in the groundwater level that affects human water accessibility, whereas the withering of plants (not as severe as the widespread death of plants) suggests a reduction in soil moisture that affects plants and crops. Consequently, the degree of drought represented by the "drying up of wells" should be considered greater than that indicated by "the grass and trees are withered and scorched."

**Line 149: Readers unfamiliar with China may not know the locations of the Wei River, Fen River, and Guanzhong Plain. The same applies to Line 189, where the locations of Changsha, Hexi Corridor, Fen and Zhang Rivers in Shanxi, Wen and Si Rivers in Shandong, Baiyangdian, Suzhou Creek, etc., have not been introduced in the article.**

Reply: Thanks for the suggestion. We decided to add another map to illustrate the geographical overview of the study area as Figure 1(b), shown below. This map has labelled the rivers and lakes mentioned in the text, such as the Wei River, Fen River, Zhang River, Dawen River, Si River, Suzhou Creek, and Baiyangdian. Additionally, we have removed or replace those terms in the manuscript that may still be less understandable to readers, such as "Guanzhong Plain," "Changsha," and "Hexi Corridor." For instance, the sentence in line 157, "the area along the Yellow, Wei, and Fen Rivers in Guanzhong Plain and southwestern Shanxi," has been revised to "the area along the Yellow, Wei, and Fen Rivers in central Shaanxi and southwestern Shanxi."

[Figure]

Figure 1: Map of the study area
(a) The location of the study area and subregions; (b)DEM of the study area with main rivers and lakes

**Figure 2a: What distinguishes a Drought Zone from a Severe Drought Zone? What criteria are used to classify areas into Drought Zones and Severe Drought Zones? Is it the case that areas experiencing level 1-2 drought events are classified as Drought Zones, while areas experiencing level 3-4 drought events are classified as Severe Drought Zones? The same question applies to the classifications in Figure 4a.**

Reply: The classification of Drought Zone and Severe Drought Zone was based on the drought kernel density distribution map for the entire study period (1627-1644). We used the natural breaks method to classify kernel densities, with densities from 0.00008 to 0.00392 representing Drought Zone, and

densities from 0.00392 to 0.02084 representing Severe Drought Zone. Thus, there is not a direct correspondence between this classification and the levels of drought events. However, the kernel density of a given raster is determined by the number and levels of drought events surrounding it.

To avoid ambiguity, we have marked the ranges of drought kernel densities corresponding to the Drought Zone and Severe Drought Zone in the legend of Figure 2a, as shown below. Similar modifications have also been made to Figure 4a.

**Figure 2b: I have the following two suggestions.**

**(1) By converting the ordinates to the same scale, the changes and differences in the DKDI index across each region can be displayed more intuitively. For example, it can be unified into a range from 0 to 1 with an interval of 0.1.**

**(2) The expression of the start and end time of each stage in the figure is different from the text content. Taking the "Starting Phase" as an example, the start time seems to be someday before 1627 and the end time seems to be someday after 1633 from the figure.**

Reply: We appreciate the suggestion to standardize the vertical axis of Figure 2b to the same scale, and range them from 0 to 1 with an interval of 0.2. Similar modifications have also been made to Figure 4b. We have also adjusted the plotting expression for phases in Figure 2b to avoid ambiguity, as shown below.

[Figure]

Figure 2: Spatial and Temporal Patterns of Drought, 1627-1644

(a) Overall spatial distribution of drought, 1627-1644. (b) Drought Kernel Density Index (DKDI) series for five regions

**Line 160-170: The changes of DKDI in the southeast and southwest regions are not mentioned in this paragraph.**

Reply: Regarding the changes of DKDI in the Southeast and Southwest regions, we have discussed this in the previous paragraph (lines 166-168). Compared to the other three regions, these two regions

have overall low DKDI values, with peaks not exceeding 0.2, indicating that they were not the main drought-affected regions. Therefore, a detailed analysis is not conducted.

**Line 171: It is recommended to provide more details on how the four phases of the drought were determined by analyzing the temporal and spatial variations of the drought.**

Reply: The four phases were determined based on the temporal changes in DKDI and the comparison of DKDI across the three main regions (Northwest, Northern, and Jianghuai regions). We have added a sentence in lines 191-193 to clarify the key events that led to the division into four phases. Text:

The delineation of the four phases is informed by key events: declining DKDI in the Northwest Region while increasing DKDI in the Northern Region in 1634; the beginning of an increase in DKDI in all three regions mainly affected in 1638; and a remarkable decline and following stabilization in DKDI in those three regions in 1642.

**Line 190: This line implies that the Yellow River once flowed through Jiangsu Province. But as we can see from Figure 1, the Yellow River does not flow through Jiangsu Province. This discrepancy is due to Figure 1 representing the modern course of the Yellow River, which has historically changed its course. This statement could potentially cause confusion among readers and therefore necessitates additional clarification.**

Reply: We have added a footnote at line 204 to explain. Text:

In the 17th century, the Yellow River flowed through Jiangsu Province into the sea. However, after a breach at Tongwagang, Henan Province, in 1855, its course changed. Currently, it does not flow through Jiangsu Province, as illustrated in Fig. 1.

**Figure 4b: Can it be divided into distinct phases like drought? If not, the phases of drought are suggested to be re-marked on the chart for easier comparison.**

Reply: To avoid confusion with the drought phases, we have not segmented the famine evolution into phases. However, we have marked the drought stages above the FKDI series graph in Fig. 4b, as shown below, to facilitate the analysis of famine evolution during different phases.

[Figure]

Figure 3: Spatial and Temporal Patterns of Famine, 1627-1644

(a) Overall spatial distribution of famine, 1627-1644. (b) Famine Kernel Density Index (FKDI) series for five regions

**Table 4: What is the correlation coefficient in the fourth year? If we want to draw conclusions that affect three years, shouldn't we at least list the correlation coefficient for the fourth year?**

Reply: Thanks for the referee's suggestion, which prompted us to revisit the correlation analysis in the original Table 4 (The correlation analysis results between DKDI and FKDI at the sub-provincial scale). We found that these results do not significantly contribute to the paper and do not provide conclusions different from those in Table 3. Therefore, we decided to remove this table and its related content. Thus, Section 5 now include: (1) correlation analysis at the regional scale to demonstrate the correlation between DKDI and FKDI, and the continuity of drought's impact on famine; (2) regression analysis at the sub-provincial scale to explore the varying contributions of drought to famine across different regions.

**Line 302: Do these 15 response types occur exclusively in the Yangtze-Huai Region, or are similar patterns observed in other regions as well?**

Reply: Measures in Table 5 were recorded across various regions, but the Jianghuai Region exhibited the greatest frequency of those measures, particularly at the local government level. In the Northern and Northwest Regions, the recorded response measures predominantly involve financial or food relief, also with some instances of giving porridge and selling grain at low prices. However, measures like treatment, corpse management, hosting of orphans, and praying for rain are rare. There are two possible reasons for this: compared to other regions, the Jianghuai Region owned a more developed economy and more abundant grain storage, providing a solid material basis for disaster response; additionally, the local officials in the Jianghuai Region showed stronger governance capabilities. Many of the measures, such as treatment and corpse management, were initiated by those officials, and the records documenting their names. It illustrates that human response measures can moderate the transmission process of drought impacts and reduce social damage. Due to space limitations, we choose the most typical region, Jianghuai Region, as an example for illustration.

**Technical corrections:**

**Line 139: Formulas need to be numbered.**

Reply: We have numbered the formulas in lines 144 and 264.

**Figure 3: The graticules appear to be missing from the figure. The same issue is observed in Figure 6.**

Reply: We have added the graticules in Figure 3 and Figure 6, as shown below.

[Figure]

Figure 3: Spatial distribution of drought during representative years of each phase

[Figure]

Figure 4: Areas affected by the locust infestation and plague

(a) Locust infestation (b)Plague

**Figure 4: Typically, figures are not placed directly beneath the title.**

Reply: We have moved Figure 4 below the first two paragraphs of Sect 4 (see line 230).

**Table 6: This table should be reformatted into a three-line table.**

Reply: We have reformatted this table into a three-line form, as shown below.

Table 5: Response measures in the Yangtze-Huai Region from 1627 to 1644

|  | Response Measure | Main Actor | Meaning |
|---|---|---|---|
| (1) | Reseeding | Farmer | After the crop dies due to drought, re-plant some if rainfall occurs. |
| (2) | Locust catching | Farmer | Catch locusts to prevent them from destroying crops. Sometimes local government also encouraged people to do so and gave them some grain or money as an award. |
| (3) | Food substitution | Famine victim | Eat wild herbs, chaff, grass roots, tree bark, soil, and so on to satisfy their hunger. The most extreme case is cannibalism (i.e. some people kill others and eat corpses). |
| (4) | Selling property | Famine victim | Selling property such as houses, land, cattle, agricultural tools, etc., in exchange for money to buy food. The most extreme case is trafficking women and children. |
| (5) | Displacement | Famine victim | Leave their hometown and flee to surrounding areas in search of food |
| (6) | Robbery | Famine victim | Rob on the road, or rob the homes of the wealthy, landlords, and gentries to obtain food or money. |
| (7) | Giving porridge | Local government, officials, gentries | Open porridge factories to feed famine victims |
| (8) | Selling grain at low price | Local government | Sell stored grain at low prices to prevent excessive increases in grain prices on the market due to shortages. |
| (9) | Financial or food relief | Local government, central government | Distribute food or money to the victims directly. |
| (10) | Donation | Officials, gentries | Donate money or grain to support measures (7)~(9). |
| (11) | Tax exemption | Central government | A discretionary exemption from taxes or corvees. |
| (12) | Tax substitution | Central government | Allow disaster areas to convert the tax grain into money or other items to hand in. |
| (13) | Treatment | Doctors, local government | Distribute medicine to the population if a plague occurred, or hire doctors to treat the patients. |
| (14) | Corpse management | Local government, officials, gentries | Distribute coffins to the families of the deceased, or bury the corpses together. |
| (15) | Hosting of orphans | Local government | Set up shelters to take in abandoned kids. |
| (16) | Praying for rain | Emperor, central officials, local officials | Offer sacrifices to gods and pray for rain. |

**Figure 7: The figure is not cited in the text.**

Reply: We have added the citation of Figure 7 in line 366.

**Referee Comment 2 (RC2)**

**1. In the discussion section, the analysis of the impact of other factors on the famine appears somewhat superficial. It could be augmented by considering the "Three Extra Levies" (special taxes imposed for military expenses – the Liao levy: imposed in response to the war with Manchuria—— later the Qing dynasty, Suppression levy: imposed in response to large-scale peasant uprisings, and Training levy: imposed for the training of new armies), which intensified the burden on peasants and influenced the famine.**

Reply: Thanks for your suggestion. We have added the "Three Extra Levies" as the fifth factor at lines 353-358. Text:

(5) The increasing taxes. Due to the Manchu invasion and peasant uprisings, the Ming government imposed additional taxes to meet the escalating military expenditures. The most notable examples were the "Three Extra Levies": the Liao levy (initiated in 1618 for the war in the northeast), the Suppression levy (initiated in 1637 to quell peasantry uprisings), and the Training levy (initiated in 1639 for training the new army). By 1639, the total amount of these additional levies even exceeded the regular tax revenue of normal years (Guo, 1983). These increasing taxes further lowered the living standards of the populace and exacerbated social conflicts.

**2. In Figure 1, important locations mentioned in the text should be marked on the map to facilitate reader comprehension. Examples include Guanzhong and the Weihe River.**

Reply: We have added another map to illustrate the geographical overview of the study area as Figure 1(b), shown below. And those important locations, especially rivers and lakes, have been marked here. At the same time, we have removed or replaced those terms in the manuscript that may still be confusing. For example, we used "central Shaanxi" to replace "Guanzhong Plain" at line 157, so that readers can refer to Figure 1(a) for the location.

[Figure]

Figure 1: Map of the study area
(a) The location of the study area and subregions; (b)DEM of the study area with main rivers and lakes

**3. In Figure 3, the first two small graphs require certain modifications. The maps for 1633 and 1636 should illustrate the average drought index during the initial phase (1627-1633) and the developmental phase (1634-1637) mentioned in the text, for clearer presentation.**

Reply: Thanks for the comment, making us think about this issue carefully. We attempted to create a map illustrating the average drought kernel density index for each phase. However, we found it to be of little help in displaying spatial variability in the drought zone. The primary reason is the differences in the spatial extent of drought from year to year, even within the same phase. Such variability would be diluted in an average kernel density map. For instance, the map below shows the average drought kernel density during the Starting Phase. It looks as if most of the study area experienced mild droughts. In reality, specific areas suffered severe droughts, such as Hebei Province in 1628 and Shaanxi and Shanxi Provinces in 1633, while most areas experienced drought only in a single year.

Therefore, we opted to maintain the original mapping approach, i.e., to present one representative year from each of the Starting, Transition, and Receding Phases, and to display the distribution of droughts for each year during the Peak Phase. Readers who wish to get a more complete picture of the spatial variability of drought are referred to the year-by-year spatial distribution maps provided in Appendix B.

[Figure]

Figure: The average drought kernel density in the starting phase (1627-1633)

**Referee Comment 3 (RC3)**

**General questions:**

**Comment 1: Insofar as famine does not affect all regions at the same time, aren't subsistence crises partly "exchange-soluble"? In other words, are there attempts (by local or central government, farmers...) to alleviate famines - on a large or small scale - through massive imports from other regions, through trade between upstream and downstream regions, or through smuggling? Similarly, is there any legislation against trade in times of crisis?**

Reply: Regional interaction plays a crucial role in addressing famine. However, during 1627-1644, regional interaction mainly manifested through the form of human migration, wherein victims fled their hometowns and migrated to surrounding areas in search of food (measure (5) in Table 5).

As for the transfer of grain, there are a few documented instances, such as in Ankang County, Shaanxi Province, where "Liu Yingke returned with rice from Hubei Province by boat and dispersed it（刘盈科自楚贩米归，倾舟散之）". However, such records are rare. Conversely, numerous records said 'no place to buy grain (无籴处)', suggesting that even those with financial means could not procure grain from other areas.

Several factors contributed to this situation: (1) The study period coincided with the end of the Ming Dynasty, characterized by a widespread shortage of grain storage across the country. Additionally, the central government's diminished governance capacity hindered long-distance food transportation on a national scale. (2) Adjacent regions often suffer from droughts and famines at the same time, especially in the peak phase, making it challenging for them to achieve self-sufficiency, let alone assist neighboring regions. (3) The Chongzhen period was marked by social unrest, with numerous peasant uprisings and famine victims turning to banditry. These groups blocked roads and looted food, causing great obstacles to the transport of grain.

Given this situation, the government did not prohibit trade but rather encouraged it. For example, the governor of Shaanxi, Sun Chuanting, ordered soldiers to clear the roads to facilitate rice purchasing and encouraged people to traffic grain to Hanzhong on their own. However, the scale and effect of such measures were limited, and overall, they contributed little to addressing the famine.

References:

Zhang, D.: A compendium of Chinese meteorological records of the last 3000 years (in Chinese), Phoenix Publishing House, Nanjing, China, 2004.

Sun, C.: Sun Chuanting's documents: Memorial to the throne on the money and grain of Hanzhong (in Chinese), Zhejiang People's Publishing House, Hangzhou, China, 1983.

**Comment 2: Table 6 (line 8) shows food distributions (also mentioned on line 306): do local governments own granaries to prevent food crises (which presupposes purchases and infrastructure) or do they legally take control of the trade and grain stored by rural communities?**

Reply: Measures (7) to (9) in Table 5 were all essentially to distribute grain to famine victims. These grains were predominantly sourced from warehousing storage, which in the Ming Dynasty included various types such as disaster-preparing granaries (预备仓), price-stabilizing granaries (常平仓), community granaries (社仓), and charity granaries (义仓). The first two types were constructed and managed by local governments, while the latter two were established and operated by civilian groups, with the government playing a supervisory role. The grain stored in these facilities was sourced from agricultural taxes, grain purchased in bountiful years, and criminal fines.

In addition to warehousing, donations played a significant role, as officials and wealthy gentry contributed money or grain during disaster years, which was then used for distributing porridge, stabilizing grain prices, or providing direct relief.

Given the crucial role of donations during the Chongzhen Drought, we decided to include it in Table 5 as measure (10) and introduce it in lines 319-320 in the manuscript. In addition, we have also added some sentences in lines 316-319 to explain the warehousing system in the Ming dynasty briefly. Text:

Measures (7) ~ (9) distributed grain to the hungry in different ways. These grains were mainly sourced from warehousing storage, which in the Ming Dynasty included various types such as disaster-preparing granaries, price-stabilizing granaries, community granaries, and charity granaries (Chen, 1991). The first two were managed by local governments, while civilian groups operated the latter two. Donations, represented by measure (10), also made a great contribution to food supply in disaster years.

Table 5: Response measures in the Yangtze-Huai Region from 1627 to 1644

| | Response Measure | Main Actor | Meaning |
|---|---|---|---|
| (10) | Donation | Local officials, gentries | Donate money or grain to support measures (7)~(9). |

References:
Chen, G.: Research on the warehouse system and famine preparation in the Ming Dynasty (in Chinese), Seeker, (5), 1991.

**Specific questions:**

**Comment 3: In table 6, line 15, "Praying for rain": is this a political and religious initiative, or just an individual and local one? Are there any general incentives that might underline the level of social stress?**

Reply: The records of praying for rain identify three types of actors: (1) the emperor, for example, "In the summer (of 1633), there was a severe drought. The emperor pardoned prisoners and prayed in the southern suburbs, after which heavy rain fell (夏大旱, 清理狱囚，上步祷南郊, 回銮, 大雨)"; (2) central officials, for example, "On the day of Gui Si in the fourth month (of 1628), the Emperor ordered the Ministry of Rites to pray for rain (四月癸巳，谕礼部祷雨)"; (3) local officials, for example, in Jiangyin County, Jiangsu Province, "There was no rain from the 5th to the 7th months (of 1640). Zhang Sijia, in charge of coastal defense, initiated prayer with his subordinates. Zhang Fenghe, the local education commissioner, also participated in the prayer (夏五月不雨至七月，海防张嗣嘉率属步祷，学台张凤翮亦出祷)."

These examples demonstrate that praying for rain was a political act. Influenced by the Confucian theory of "interactions between heaven and mankind", the government intended to show concern for droughts and self-reflection by praying. This practice helped to appease the populace and mitigate tensions to some extent. However, in the case of Chongzhen Drought, the praying records are sparse and lack detailed descriptions of cause and effect, making further analysis difficult.

We have supplemented the "main actor" field of measure (16)-praying for rain in Table 5 to indicate its attributes of political act. As follows:

Table 5: Response measures in the Yangtze-Huai Region from 1627 to 1644

| | Response Measure | Main Actor | Meaning |
|---|---|---|---|
| (16) | Praying for rain | Emperor, central officials, local officials | Offer sacrifices to gods and pray for rain. |

**Comment 4: The authors mentioned "cannibalism" in Table 2 and Table 6 and state at line 317 that "Severe famine swept through the region, and there were even instances of cannibalism." Are these extreme cases sufficiently well-documented and numerous to be used in the classification scale of famine levels and response measures, or are they local epiphenomena?**

Reply: From 1627 to 1644, there were 557 records of "cannibalism", which is a large number. Over 80% of these events took place during the peak phase of drought (1638-1641). Spatially, such events happened over a wide range, but 88% concentrated in the Northwest and Northern Regions.

Cannibalism signifies that the famine has reached an extreme level of severity, reflecting both the extreme scarcity of food and the collapse of ethics and social order. Based on the records, it is evident that cannibalism, as the most extreme form of food substitution, occurred during the study period. The quantity and content of these records support their use as one criterion for classifying famine levels. However, we also believe that such extreme events should be treated with great caution. We only took them as evidence of famine without conducting further specific analyses of themselves.

**Comment 5: As RC2 pointed out, important locations mentioned in the text should be marked on the map (Figure 1) to facilitate reader comprehension.**

Reply: Thanks a lot for your comment, this is something we overlooked while writing the paper. We have redesigned Figure 1 to better present the geographical overview of the study area, as shown below. And those important locations have been marked there. At the same time, we have removed or replaced those terms in the manuscript that may still be confusing. For example, we use "central Shaanxi" to replace "Guanzhong" at line 157, so that readers can refer to Figure 1(a) for the location.

[Figure]

Figure 1: Map of the study area
(a) The location of the study area and subregions; (b) DEM of the study area with main rivers and lakes

**Editor Comment**

**In my opinion, it would also be very beneficial if you could add some of the explanations from your replies – e.g. the reply to referee report 3 – into the manuscript (where you feel it makes sense). The descriptions of the storage system, for example, are very exciting and new for many readers because other systems were used in Europe in the same period.**

Reply: Thanks a lot for your careful editing work and suggestions. We are glad that you find the description of the storage system interesting. We have added some sentences in lines 316-319 to introduce this system briefly. Text:

These grains were mainly sourced from warehousing storage, which in the Ming Dynasty included various types such as disaster-preparing granaries, price-stabilizing granaries, community granaries, and charity granaries (Chen, 1991a). The first two were managed by local governments, while civilian groups operated the latter two.

**The positions of all relevant changes in the manuscript**

Lines 40-53
Line 55
Figure 1(b)
Lines 98-101
Lines 119-120
Table 1
Line 144
Line 157
Figure 2(a)(b)
Lines 166-168
Lines 191-193
Figure 3
Line 204
Footnote 1 on page 8
Figure 4(a)(b)
Line 264
Lines 316-320
Table 5
Lines 353-358
Figure 6
Line 366

**Newly added references**

Lines 432-434
Lines 440-443
Line 454
Line 467
Lines 468-470
Line 490
Lines 499-500
Lines 505-506
Lines 530-531